# On Characterizing and Mitigating Imbalances in Multi-Instance Partial Label Learning

## Abstract

*Multi-Instance Partial Label Learning* (MI-PLL) is a weakly-supervised learning setting encompassing *partial label learning*, *latent structural learning*, and *neurosymbolic learning*. Unlike supervised learning, in MI-PLL, the inputs to the classifiers at training-time are tuples of instances $\mathbf{x}$. At the same time, the supervision signal is generated by a function $\sigma$ over the (hidden) gold labels of $\mathbf{x}$. In this work, we make multiple contributions towards addressing a problem that hasn't been studied so far in the context of MI-PLL: that of characterizing and mitigating *learning imbalances*, i.e., major differences in the errors occurring when classifying instances of different classes (aka *class-specific risks*). In terms of theory, we derive class-specific risk bounds for MI-PLL, while making minimal assumptions. Our theory reveals a unique phenomenon: that $\sigma$ can greatly impact learning imbalances. This result is in sharp contrast with previous research on supervised and weakly-supervised learning, which only studies learning imbalances under the prism of data imbalances. On the practical side, we introduce a technique for estimating the marginal of the hidden labels using only MI-PLL data. Then, we introduce algorithms that mitigate imbalances at training- and testing-time, by treating the marginal of the hidden labels as a constraint. We demonstrate the effectiveness of our techniques using strong baselines from neurosymbolic and long-tail learning, suggesting performance improvements of up to 14%.

## 1 Introduction

The need to reduce labeling costs motivates the study of weakly-supervised learning settings (Zhou, 2017; Zhang et al., 2022). Our work aligns with this objective, focusing on *multi-instance partial label learning* (MI-PLL) (Wang et al., 2023b). MI-PLL is particularly appealing, as it encompasses three well-known learning settings: *partial label learning* (PLL) (Cour et al., 2011; Cabannes et al., 2020; Lv et al., 2020; Seo & Huh, 2021; Wen et al., 2021; Xu et al., 2021; Yu et al., 2022; Wang et al., 2022; Hong et al., 2023), where each training instance is associated with a set of candidate labels, *latent structural learning* (Steinhardt & Liang, 2015; Raghunathan et al., 2016; Zhang et al., 2020), i.e., learning classifiers subject to a transition function $\sigma$ that constraints their outputs, and *neurosymbolic learning* (Manhaeve et al., 2018; Wang et al., 2019b; Dai et al., 2019; Tsamoura et al., 2021; Huang et al., 2021; Li et al., 2023a), i.e., training neural classifiers subject to symbolic background knowledge. An example (adapted from (Manhaeve et al., 2018)) is illustrated below:

**Example 1.1** (MI-PLL example). *We aim to learn an MNIST classifier $f$, using only samples of the form $(x_1, x_2, s)$, where $x_1$ and $x_2$ are MNIST digits and $s$ is the maximum of their gold labels, i.e., $s = \sigma(y_1, y_2) = \max\{y_1, y_2\}$ with $y_i$ being the label of $x_i$. The gold labels are hidden during training. We will refer to the $y_i$'s and $s$ as* hidden *and* partial *labels, respectively.*

MI-PLL has been a topic of active research in NLP (Steinhardt & Liang, 2015; Raghunathan et al., 2016; Peng et al., 2018; Wang et al., 2019a; Gupta et al., 2021). Recently, it has received renewed attention in neurosymbolic learning, as it offers multiple benefits over architectures that approximate the neural classifiers and $\sigma$ via end-to-end neural models, such as (i) the ability to reuse the latent models (Peng et al., 2018; Mihaylova et al., 2020), (ii) higher accuracy (Wu, 2022; Huang et al., 2021), and (iii) higher explainability and generalizability. Practical applications of MI-PLL in the neurosymbolic learning literature include visual question answering (Huang et al., 2021), video-text retrieval (Li et al., 2023b), and fine-tuning language models (Zhang et al., 2023; Li et al., 2024).

For the first time, we address an unexplored topic in the context of MI-PLL: that of characterizing and mitigating *learning imbalances*, i.e., major differences in the errors occurring when classifying instances of different classes (aka *class-specific risks*).

Existing works in supervised (Menon et al., 2021; Cao et al., 2019) and weakly-supervised learning (Wang et al., 2022; Hong et al., 2023) study imbalances under the prism of *long-tailed* (aka *imbalanced*) data: data in which instances of different classes occur with very different frequencies, (He & Garcia, 2009; Horn & Perona, 2017; Buda et al., 2018). However, those results cannot characterize learning imbalances in MI-PLL. This is because *transition function σ may cause learning imbalances even when the hidden or the partial labels are uniformly distributed*. Fig-

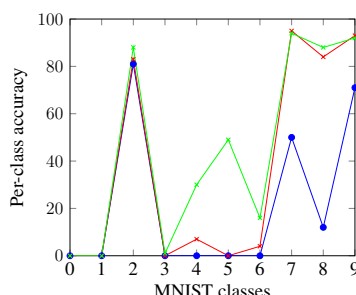

Figure 1: Accuracy of the classifier from Example 1.1. Blue, red and green curves show accuracy at 20, 40 and 100 epochs. Learning converges in 100 epochs.

ure 1 demonstrates this phenomenon by showing the per-class classification accuracy across different training epochs when an MNIST classifier is trained as in Example 1.1 and the hidden labels are uniform. Hence, to formally characterise imbalances in MI-PLL, we need to account for $\sigma$.

On the practical side, mitigating learning imbalances has received considerable attention in supervised and weakly-supervised learning with the proposed techniques (typically referred to as *long-tail learning*) operating at training- (Cao et al., 2019; Tan et al., 2020; 2021; Chawla et al., 2002; Buda et al., 2018) or at testing-time (Kang et al., 2020; Peng et al., 2022; Menon et al., 2021).

However, there are two main reasons that make previous practical algorithms on long-tail leaning not appropriate for MI-PLL. First, they rely on (good) approximations of the marginal distribution of the hidden labels. While approximating **r** may be easy in supervised learning (Menon et al., 2021) as the gold labels are available, in our setting the gold labels are hidden from the learner. Second, the state-of-the-art for training-time mitigation (Wang et al., 2022; Cao et al., 2019; Tan et al., 2020; 2021; Chawla et al., 2002; Buda et al., 2018; Hong et al., 2023) is designed for settings in which a single instance is presented each time to the learner and hence, they cannot take into account the correlations among the instances. The above gives rise to a second challenge: *developing techniques for mitigating learning imbalances in MI-PLL*.

**Contributions.** We start by providing class-specific error bounds in the context of MI-PLL. Complementary to previous work in supervised learning (Cao et al., 2019) and standard single-instance PLL (Cour et al., 2011), our theory shows that $\sigma$ can have a significant impact on learning imbalances, see Theorem 3.1. Our analysis extends the theoretical analysis in (Wang et al., 2023b), by providing stricter risk bounds for the underlying classifiers, making also minimal assumptions, and the theoretical analysis in (Cour et al., 2011) that provides class-specific error bounds for standard PLL.

On the practical side, we first propose a statistically consistent technique for estimating the marginal of the hidden labels given partial labels. We further propose two algorithms that mitigate imbalances at training- and testing-time. The first algorithm assigns pseudo-labels to training data based on a novel linear programming formulation of MI-PLL, see Section 4.2. The second algorithm uses the hidden label marginals to constrain the model's prediction on testing data, using a robust semi-constrained optimal transport (RSOT) formulation (Le et al., 2021), see Section 4.3. Our empirical analysis shows that our techniques can improve the accuracy over strong baselines in neurosymbolic learning (Xu et al., 2018; Wang et al., 2023b) and long-tail learning (Menon et al., 2021; Hong et al., 2023) by up to 14%, manifesting that the straightforward application of state-of-the-art to MI-PLL settings is either impossible (Wang et al., 2022) or problematic (Hong et al., 2023).

## 2 PRELIMINARIES

Our notation is summarized in Table 7 and 8 and builds upon (Wang et al., 2023b).

**Data and models.** For an integer $n \geq 1$, let $[n] := \{1, \ldots, n\}$. Let also $\mathcal{X}$ be the instance space and $\mathcal{Y} = [c]$ be the output space. We use $x, y$ to denote elements in $\mathcal{X}$ and $\mathcal{Y}$. The joint distribution of two random variables $X, Y$ over $\mathcal{X} \times \mathcal{Y}$ is denoted as $\mathcal{D}$, with $\mathcal{D}_X, \mathcal{D}_Y$ denoting marginals of $X$ and $Y$. Vector $\mathbf{r} = (r_1, \ldots, r_c)$ denotes $\mathcal{D}_Y$, where $r_j := \mathbb{P}(Y = j)$ is the probability of occurrence (or

ratio) of label $j \in \mathcal{Y}$ in $\mathcal{D}$. We consider *scoring functions* of the form $f : \mathcal{X} \to \Delta_c$, where $\Delta_c$ is the space of probability distributions over $\mathcal{Y}$, e.g., $f$ outputs the softmax probabilities (or *scores*) of a neural classifier. We use $f^j(x)$ to denote the score of $f(x)$ for class $j \in \mathcal{Y}$. A scoring function $f$ induces a *classifier* $[f] : \mathcal{X} \to \mathcal{Y}$, whose *prediction* on $x$ is given by $\operatorname{argmax}_{j \in [c]} f^j(x)$. We denote by $\mathcal{F}$ the set of scoring functions and by $[\mathcal{F}]$ the set of induced classifiers. The *zero-one loss* is given by $L(y', y) := \mathbb{1}\{y' \neq y\}$. The *zero-one risk* of $f$ is given by $R(f) := \mathbb{E}_{(X,Y) \sim \mathcal{D}}[L([f](X), Y)]$. The risk of $f$ for class $j$ is defined as the probability of $f$ mispredicting an instance of that class, i.e., $R_j(f) := \mathbb{P}([f](x) \neq j | Y = j)$. We refer to that risk as the *class-specific* one.

**Multi-Instance PLL.** We set $\mathbf{x} = (x_1, \ldots, x_M)$ and denote by $\mathbf{y} = (y_1, \ldots, y_M)$ the corresponding gold labels. Let $\sigma : \mathcal{Y}^M \to \mathcal{S}$ be a transition function. Space $\mathcal{S} = \{a_1, \ldots, a_{c_S}\}$ is referred to as the *partial label space*, where $|\mathcal{S}| = c_S \geq 1$. We assume that $\sigma$ is known to the learner, a common assumption in neurosymbolic learning (Dai et al., 2019; Li et al., 2023a). Let $\mathcal{T}_\mathsf{P}$ be a set of $m_\mathsf{P}$ *partially labeled* samples of the form $(\mathbf{x}, s) = (x_1, \ldots, x_M, s)$. We refer to $s$ as a *partial label*. Each partially labeled sample is formed by drawing $M$ i.i.d. samples $(x_i, y_i)$ from $\mathcal{D}$ and setting $s =: \sigma(y_1, \ldots, y_M)$. The distribution of samples $(\mathbf{x}, s)$ is denoted by $\mathcal{D}_\mathsf{P}$. We set $[f](\mathbf{x}) := ([f](x_1), \ldots, [f](x_M))$. The *zero-one partial loss subject to $\sigma$* is defined as $L_\sigma(\mathbf{y}, s) := L(\sigma(\mathbf{y}), s) = \mathbb{1}\{\sigma(\mathbf{y}) \neq s\}$, for any $\mathbf{y} \in \mathcal{Y}^M$ and $s \in \mathcal{S}$. Learning aims to finding the classifier $f$ with the minimum *zero-one partial risk subject to $\sigma$* given by $R_\mathsf{P}(f; \sigma) := \mathbb{E}_{(X_1, \ldots, X_M, S) \sim \mathcal{D}_\mathsf{P}}[L_\sigma(([f](\mathbf{X})), S)]$.

**Vectors and matrices.** A vector $\mathbf{v}$ is *diagonal* if all of its elements are equal. We denote by $\mathbf{e}_i$ the one-hot vector, where the $i$-th element equals to 1. We denote the all-one and all-zero vectors by $\mathbf{1}_n$ and $\mathbf{0}_n$, and the identity matrix of size $n \times n$ by $\mathbf{I}_n$. Let $\mathbf{A} \in \mathbb{R}^{n \times m}$ be a matrix. We use $A_{i,j}$ to denote the value of the $(i, j)$ cell of $\mathbf{A}$ and $v_i$ to denote the $i$-th element of $\mathbf{v}$. The *vectorization* of $\mathbf{A}$ is given by $\operatorname{vec}(\mathbf{A}) := [a_{1,1}, \ldots, a_{n,1}, \ldots, a_{1,m}, \ldots, a_{n,m}]^\mathsf{T}$ and its *Moore–Penrose inverse* by $\mathbf{A}^\dagger$. If $\mathbf{A}$ is square, then the diagonal matrix that shares the same diagonal with $\mathbf{A}$ is denoted by $D(\mathbf{A})$. For matrices $\mathbf{A}$ and $\mathbf{B}$, $\mathbf{A} \otimes \mathbf{B}$ and $\langle \mathbf{A}, \mathbf{B} \rangle$ denote their *Kronecker* and *Frobenius inner products*.

## 3 Theory: characterizing learning imbalances in MI-PLL

This section theoretically characterizes learning imbalances in MI-PLL by providing class-specific risk bounds, see Proposition 3.1. These bounds measure the difficulty of learning instances of each class in $\mathcal{Y}$, indicating that, unlike supervised learning, *learning imbalances in MI-PLL arise not only from label distribution imbalances but also from the partial labeling process $\sigma$*. Unlike prior work (Wang et al., 2023b), our analysis relies solely on the i.i.d. assumption (see Section 2). To ease the presentation, we focus on $M = 2$. Nevertheless, our analysis directly generalizes for $M > 2$.

Our theory is based on a novel non-linear program formulation that allows us to compute an upper bound of each $R_j(f)$. The first key idea (K1) to that formulation is a rewriting of $R_\mathsf{P}(f; \sigma)$ and $R_j(f)$. To start with, given the transition $\sigma$, the zero-one partial risk can be expressed as

probability of the label pair $(i, j)$          the partial label is misclassified

$$R_\mathsf{P}(f; \sigma) = \sum_{(i,j) \in \mathcal{Y}^2} r_i r_j \left( \sum_{(i',j') \in \mathcal{Y}^2} \mathbb{1}\{\sigma(i,j) \neq \sigma(i',j')\} \, \mathbf{H}_{ii'}(f) \mathbf{H}_{jj'}(f) \right) \tag{1}$$

conditional probability that the labels $i$ and $j$ are (mis)classified as $i'$ and $j'$

where $\mathbf{H}(f)$ is an $c \times c$ matrix defined as $\mathbf{H}(f) := [\mathbb{P}([f](x) = j | Y = i)]_{i \in [c], j \in [c]}$. Equation (1) is a straightforward rewriting of $R_\mathsf{P}(f; \sigma)$, see Section 2. To derive (1), we enumerate all the 4-ary vectors $(i, j, i', j') \in \mathcal{Y}^4$, where $i, j$ are the gold hidden labels and $i', j'$ are the predicted labels, so that the predicted labels lead to a wrong partial label, i.e., $\sigma(i, j) \neq \sigma(i', j')$. The risk $R_\mathsf{P}(f; \sigma)$ is the sum of the probabilities of those wrong predictions, with $H_{ii'}(f) H_{jj'}(f)$ encoding the probability of occurrence of the vectors $(i, j, i', j')$. Now, let $\mathbf{h}(f) = \operatorname{vec}(\mathbf{H}(f))$ be the vectorization of $\mathbf{H}(f)$. The partial risk $R_\mathsf{P}(f; \sigma)$ in (1) is a quadratic form of $\mathbf{h}(f)$. Therefore, there is a unique symmetric matrix $\mathbf{\Sigma}_{\sigma, \mathbf{r}}$ in $\mathbb{R}^{c^2 \times c^2}$ that depends only on $\sigma$ and $\mathbf{r}$ such that (1) can be rewritten as $R_\mathsf{P}(f; \sigma) = \mathbf{h}(f)^\mathsf{T} \mathbf{\Sigma}_{\sigma, \mathbf{r}} \mathbf{h}(f)$. Furthermore, for each $j \in \mathcal{Y}$, let $\mathbf{W}_j$ be the matrix defined by $(\mathbf{1}_c - \mathbf{e}_j)\mathbf{e}_j^\mathsf{T}$ and $\mathbf{w}_j$ be its vectorization. We can rewrite the class-specific risk as

$$R_j(f) = \mathbf{w}_j^\mathsf{T} \mathbf{h}(f) \tag{2}$$

The second key idea (K2) to forming a non-linear program for computing class-specific risk bounds is to upper bound the class-specific risk $R_j(f)$ of a model $f$ with the model's partial risk $R_\mathsf{P}(f; \sigma)$.

The latter can be minimized with partially labeled data $\mathcal{T}_\mathsf{P}$. Putting (K1) and (K2) together, the worst class-specific risk of $f$ for class $j \in \mathcal{Y}$ is given by the optimal solution to the program below:

$$
\begin{aligned}
\max_{\mathbf{h}} \quad & \mathbf{w}_j^\mathsf{T} \mathbf{h}(f) \\
\text{s.t.} \quad & \mathbf{h}(f)^\mathsf{T} \mathbf{\Sigma}_{\sigma,\mathbf{r}} \mathbf{h}(f) = R_\mathsf{P}(f;\sigma) && \text{(partial risk)} \\
& \mathbf{h}(f) \geq 0 && \text{(positivity)} \\
& (\mathbf{I}_c \otimes \mathbf{1}_c^\mathsf{T})\mathbf{h}(f) = \mathbf{1}_c && \text{(normalization)}
\end{aligned}
\tag{3}
$$

Let's analyze (3). The optimization objective states that we aim to find the worst possible class-specific risk as expressed in (2). The first constraint specifies the partial risk of the model. The second one asks the (mis)classification probabilities to be non-negative. The last constraint, where $(\mathbf{I}_c \otimes \mathbf{1}_c^\mathsf{T})\mathbf{h}(f)$ represents the row sums of matrix $\mathbf{H}(f)$, requires the classification probabilities to sum to one. Let $\Phi_{\sigma,j}(R_\mathsf{P}(f;\sigma))$ denote the optimal solution to program (3). Formally, we have:

**Proposition 3.1** (Class-specific risk bound). *For any $j \in \mathcal{Y}$, we have that $R_j(f) \leq \Phi_{\sigma,j}(R_\mathsf{P}(f;\sigma))$.*

**Characterizing learning imbalance.** Proposition 3.1 suggests that the worst risk associated with each class in $\mathcal{Y}$ is characterized by two factors. The first one is the model's partial risk $R_\mathsf{P}(f;\sigma)$, which is independent of the specific class. The second factor is $\sigma$, as $\sigma$ impacts on the mapping $\Phi_{\sigma,j}$ from the model's partial risk to the class-specific risk. Therefore, the learning imbalance can be assessed by comparing the growth rates of $\Phi_{\sigma,j}$. We use this approach below to analyze Example 1.1.

**Example 3.2** (Cont' Example 1.1). *Let $\mathcal{D}$ and $\mathcal{D}_\mathsf{P}$ be defined as in Section 2. Consider the two cases:*

CASE 1 *The marginal of the hidden label $Y$ is uniform. The left-hand side of Figure 2 shows the risk bounds for different classes obtained via solving program (3). The bounds are presented as functions of different values of $R_\mathsf{P}(f;\sigma)$. In this plot, the curve for class "zero" (resp. "nine") has the steepest (resp. smoothest) slope, suggesting that $f$ will tend to make more (resp. fewer) mistakes when classifying instances of that class. In other words, class "zero" is the hardest to learn, as also shown to be the case in reality, see Figure 1.*

CASE 2 *The marginal of the partial label $S$ is uniform. Similarly, the right-hand side Figure 2 plots the corresponding risk bounds, suggesting that the class "zero" is now the easiest to learn.*

**Obtaining the label ratio $\mathbf{r}$.** Computing the program (3) requires knowing the transition $\sigma$ and the label distribution $\mathbf{r}$. While $\sigma$ is assumed to be given, $\mathbf{r}$ may be unknown in practice. To circumvent this, in Section 4.1, we present a technique for estimating $\mathbf{r}$ using only partially labeled data $\mathcal{T}_\mathsf{P}$.

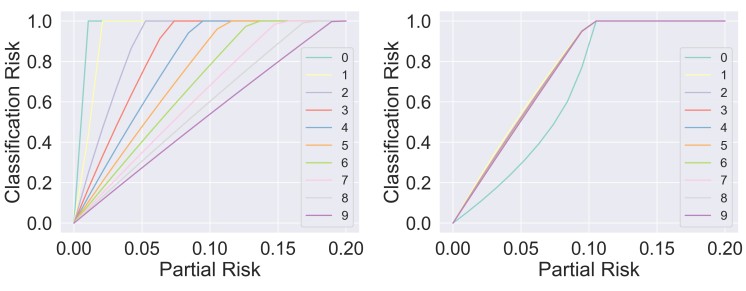

Figure 2: Class-specific upper bounds obtained via (3). (left) $\mathcal{D}_Y$ is uniform. (right) $\mathcal{D}_{\mathsf{P}_S}$ is uniform.

**Computable bounds for $R_j(f)$.** Via Proposition 3.1, we could further derive a bound for $R_j(f)$ that can be computed using an MI-PLL dataset. This can be done by using standard learning theory tools (e.g., VC-dimension or Rademacher complexity) to show that, given a fixed confidence level $\delta \in (0,1)$, the partial risk $R_\mathsf{P}(f;\sigma)$ will not exceed a *generalization bound* $\widetilde{R}_\mathsf{P}(f;\sigma,\mathcal{T}_\mathsf{P},\delta)$ with probability $1 - \delta$. An example is shown below.

**Proposition 3.3.** *Let $d_{[\mathcal{F}]}$ be the Natarajan dimension of $[\mathcal{F}]$. Given a confidence level $\delta \in (0,1)$, we have that $R_j(f) \leq \Phi_{\sigma,j}(\widetilde{R}_\mathsf{P}(f;\sigma,\mathcal{T}_\mathsf{P},\delta))$ with probability $1 - \delta$ for any $j \in [c]$, where*

$$
\widetilde{R}_\mathsf{P}(f;\sigma,\mathcal{T}_\mathsf{P},\delta) = \widehat{R}_\mathsf{P}(f;\sigma,\mathcal{T}_\mathsf{P}) + \sqrt{\frac{2\log(em_\mathsf{P}/2d_{[\mathcal{F}]}\log(6Mc^2 d_{[\mathcal{F}]}/e))}{m_\mathsf{P}/2d_{[\mathcal{F}]}\log(6Mc^2 d_{[\mathcal{F}]}/e)}} + \sqrt{\frac{\log(1/\delta)}{2m_\mathsf{P}}}
\tag{4}
$$

The first term in the right-hand side of (4) denotes the empirical partial risk of classifier $f$, the second one upper bounds the Natarajan dimension of $f$ (Shalev-Shwartz & Ben-David, 2014), and the third

term quantifies the confidence level or the probability that the generalization bound holds, which is typical in learning theory. Proposition 3.3 shows how fast the risk of $f$ for class $j \in \mathcal{Y}$ decreases when training using partial labels. A further discussion on our bounds and Example 3.2 is in B.2.

**Comparison to previous work.** The most relevant work to ours is (Wang et al., 2023b), which first establishes the learnability for MI-PLL. Our result extends (Wang et al., 2023b) in three ways: (i) we bound the class-specific risks $R_j(f)$ instead of bounding the total risk $R(f)$; (ii) our bounds do not rely on $M$-unambiguity, in contrast to those in (Wang et al., 2023b); and (iii) the program (3) leads to tighter bounds for $R(f)$. Before proving (iii), let us first recapitulate $M$-unambiguity:

**Definition 3.4** ($M$-unambiguity from (Wang et al., 2023b)). *A transition $\sigma$ is $M$-unambiguous if for any two diagonal label vectors $\mathbf{y}$ and $\mathbf{y}' \in \mathcal{Y}^M$ such that $\mathbf{y} \neq \mathbf{y}'$, we have that $\sigma(\mathbf{y}') \neq \sigma(\mathbf{y})$.*

Let us illustrate (iii) from above. By relaxing the constraints in (3), we can recover Lemma 1 from (Wang et al., 2023b) (which is the key to proving Theorem 1 from (Wang et al., 2023b)). In particular, if we: (1) drop the the positivity and normalization constraints from (3) and (2) replace the partial risk constraint by a more relaxed inequality $\mathbf{h}(f)^\mathsf{T} D(\mathbf{\Sigma}_{\sigma,\mathbf{r}}) \mathbf{h}(f) \leq R_\mathsf{P}(f; \sigma)$, we obtain the following:

**Proposition 3.5.** *If $\sigma$ is $M$-unambiguous, then the risk of $f$ can be bounded by*

$$R(f) \leq \sqrt{\mathbf{w}^\mathsf{T} (D(\mathbf{\Sigma}_{\sigma,\mathbf{r}}))^\dagger \mathbf{w} R_\mathsf{P}(f; \sigma)} = \sqrt{c(c-1) R_\mathsf{P}(f; \sigma)} \tag{5}$$

*which coincides with Lemma 1 from (Wang et al., 2023b) for $M = 2$, where $\mathbf{w} := \sum_{j=1}^c r_j \mathbf{w}_j$.*

# 4 ALGORITHMS: MITIGATING IMBALANCES IN MI-PLL

Section 3 sends a clear message: MI-PLL is prone to learning imbalances that may be exacerbated due to $\sigma$. We now propose a portfolio of techniques for addressing learning imbalances. Our first contribution, see Section 4.1, is a statistically consistent technique for estimating $\mathbf{r}$, assuming access to partial labels only. We then move to training-time mitigation, see Section 4.2 and testing-time mitigation, see Section 4.3. Our marginal estimation algorithm requires only the i.i.d. assumption; the algorithms in Section 4.2 and 4.3 work even when the i.i.d. assumption fails. Our mitigation algorithms enforce the class priors to a classifier's predictions. This is a common idea in long-tail learning. The intuition is that the classifier will tend to predict the labels that appear more often in the training data. Enforcing the priors, gives more importance to the minority classes at training-time (see Section 4.2) and encourages the model to predict minority classes at testing-time (see Section 4.3).

## 4.1 ESTIMATING THE MARGINAL OF THE HIDDEN LABELS

We begin with our technique for estimating $\mathbf{r}$ using only partially labeled data $\mathcal{T}_\mathsf{P}$. Let us first introduce our notation. We denote the probability of occurrence (or ratio) of the $j$-th partial label $a_j \in \mathcal{S}$ by $p_j := \mathbb{P}(S = a_j)$ and set $\mathbf{p} = (p_1, \ldots, p_{c_S})$. We also denote the set of all label vectors that map to $s$ under $\sigma$ by $\sigma^{-1}(s)$. In terms of Example 1.1, $\sigma^{-1}(s = 1) = \{(0,1),(1,0),(1,1)\}$. To estimate $\mathbf{r}$, we rely on the observation that in MI-PLL, $p_j$ equals the probability of the label vectors in $\sigma^{-1}(a_j)$, namely $p_j = \sum_{(y_1,\ldots,y_M) \in \sigma^{-1}(a_j)} \prod_{i=1}^M r_{y_i}$, which is a polynomial of $\mathbf{r}$. We use $P_\sigma$ to refer to the system of polynomial equations $[p_j]_{j \in [c_S]}^\mathsf{T} = [\sum_{(y_1,\ldots,y_M) \in \sigma^{-1}(a_j)}]_{j \in [c_S]}^\mathsf{T}$.

**Example 4.1.** *Consider* CASE (2) *from Example 3.2. Assume that the marginals of the partial labels are uniform. Then, we can obtain $\mathbf{r}$ via solving the following system of polynomial equations: $[r_0^2, r_1^2 + 2r_0 r_1, \ldots, r_9^2 + 2\sum_{i=0}^8 r_i r_9]^\mathsf{T} = [1/10, 1/10, \ldots, 1/10]^\mathsf{T}$. The first equation denotes the probability a partial label to be zero, which is $1/10$ (uniformity). Due to $\sigma$, this can happen only when $y_1 = y_2 = 0$. Under the independence assumption, the above implies that $r_0^2 = 1/10$. Analogously, the second and the last polynomials denote the probabilities a partial label to be one and nine.*

Let $\Psi_\sigma$ be the function mapping each $r_j \in \mathcal{Y}$ to its solution in $P_\sigma$, assuming $\mathbf{p}$ is known. In practice, $\mathbf{p}$ is unknown, but can be estimated by the empirical distribution of a partially labeled dataset $\mathcal{T}_\mathsf{P}$ of size $m_\mathsf{P}$, namely $\bar{p}_j := \sum_{k=1}^{m_\mathsf{P}} \mathbb{1}\{s_k = a_j\}/m_\mathsf{P}$. As the $\bar{p}_j$'s can be noisy, the system of polynomials could become inconsistent. Therefore, instead of solving the polynomial equation as in Example 4.1, we find an estimate $\widehat{\mathbf{r}}$, so that its induced prediction for the partial label ratio $\widehat{\mathbf{p}} := \Psi_\sigma(\widehat{\mathbf{r}})$ best fits to the empirical probabilities $\bar{p}_j$'s by means of cross-entropy. Since this requires optimizing over the probability simplex $\Delta_c$, we reparametrize the estimated ratios $\widehat{\mathbf{r}}$ by $\text{softmax}(\mathbf{u})$, leading to Algorithm 1. We provide a theoretical guarantee for the consistency of Algorithm 1 in Appendix C.

| **Algorithm 1** LABEL RATIO SOLVER | **Algorithm 2** CAROT |
|---|---|
| **Input:** partial labels $\{s_k\}_{k=1}^{m_P}$, transition function $\sigma$, step size $t$, iterations $N_{\text{iter}}$ | **Input:** model's raw scores $\mathbf{P} \in \mathbb{R}^{c \times n}$, ratio estimates $\widehat{\mathbf{r}} \in \mathbb{R}^c$, entropic reg. parameter $\eta > 0$, margin reg. parameter $\tau > 0$, iterations $N_{\text{iter}}$ |
| **Initialize:** logit $\mathbf{u} \leftarrow \mathbf{1}_c$; $\bar{p}_j$, for $j \in [c_S]$ | **Initialize:** $\mathbf{u} \leftarrow \mathbf{0}_n$; $\mathbf{v} \leftarrow \mathbf{0}_c$ |
| **for** $N = 1, \ldots, N_{\text{iter}}$ **do** | **for** $N = 1, \ldots, N_{\text{iter}}$ **do** |
| $\quad \widehat{\mathbf{r}} \leftarrow \text{softmax}(\mathbf{u})$ | $\quad \mathbf{a} \leftarrow B(\mathbf{u}, \mathbf{v})\mathbf{1}_c; \quad \mathbf{b} \leftarrow B(\mathbf{u}, \mathbf{v})^\mathsf{T} \mathbf{1}_n$ |
| $\quad$ **for each** $j \in [c_S]$ **do** | $\quad$ **if** $k$ is even **then** |
| $\quad\quad \widehat{p}_j \leftarrow \sum\limits_{(y_1, \ldots, y_M) \in \sigma^{-1}(a_j)} \prod_{i=1}^M \widehat{r}_{y_i}$ | $\quad\quad$ **update v** //see Section 4.3 |
| $\quad \ell \leftarrow \sum_{j=1}^{c_S} \bar{p}_j \log \widehat{p}_j$ | $\quad$ **else** |
| $\quad$ Backpropagate $\ell$ to update $\mathbf{u}$ | $\quad\quad$ **update u** //see Section 4.3 |
| **return** $\text{softmax}(\mathbf{u})$ | **return** $B(\mathbf{u}, \mathbf{v})$ |

## 4.2 TRAINING-TIME IMBALANCE MITIGATION VIA LINEAR PROGRAMMING

We now turn to training-time mitigation. We aim to find pseudo-labels $\mathbf{Q}$ that are close to the classifier's scores and adhere to $\widehat{\mathbf{r}}$ and use $\mathbf{Q}$ to train the classifier using the cross-entropy loss. There are two design choices: (i) whether to find pseudo-labels at the individual instance level or at the batch level; (ii) whether to be strict in enforcing the marginal $\widehat{\mathbf{r}}$. In addition, we face two challenges: (iii) we are provided with $M$-ary tuples of instances of the form $(x_1, \ldots, x_M)$; (iv) $\mathbf{Q}$ must additionally abide by the constraints coming from $\sigma$ and the partial labels, e.g., when $s = 1$ in Example 1.1, then the only valid label assignments for $(x_1, x_2)$ are (1,1), (0,1) and (1,0). Regarding (i), finding pseudo-labels at the individual instance level does not guarantee that the modified scores match $\widehat{\mathbf{r}}$ (Peng et al., 2022). Regarding (ii), strictly enforcing $\widehat{\mathbf{r}}$ could be problematic as $\widehat{\mathbf{r}}$ can be noisy.

To accommodate the above requirements while avoiding the crux of solving non-linear programs, we rely on a novel *linear programming* (LP) formulation of MI-PLL that finds pseudo-labels for a batch of $n$ scores. We use $(x_{\ell,1}, \ldots, x_{\ell,M}, s_\ell)$ to denote the $\ell$-th partial training sample in a batch of size $n$. We also use $\mathbf{P}_i \in [0,1]^{n \times c}$ and $\mathbf{Q}_i \in [0,1]^{n \times c}$, for $i \in [M]$, to denote the classifier's scores and the pseudo-labels assigned to the $i$-th input instances of the batch. In particular, $P_i[\ell, j] = f^j(x_{\ell,i})$, while $Q_i[\ell, j]$ is the corresponding pseudo-label. Before continuing, it is crucial to explain how to associate each training sample $s_\ell$ with a Boolean formula in *disjunctive normal form* (DNF). Associating partial labels with DNF formulas is standard in the neurosymbolic literature (Xu et al., 2018; Tsamoura et al., 2021; Huang et al., 2021; Wang et al., 2023b). For $\ell \in [n]$, $i \in [M]$, and $j \in [c]$, let $q_{\ell,i,j}$ be a Boolean variable that is true if $x_{\ell,i}$ is assigned label $j \in \mathcal{Y}$ and false otherwise. Let $R_\ell$ be the size of $\sigma^{-1}(s_\ell)$. Based on the above, we can associate each label vector $\mathbf{y}$ in $\sigma^{-1}(s_\ell)$ with a conjunction $\phi_{\ell,t}$ of Boolean variables from $\{q_{\ell,i,j}\}_{i \in [M], j \in [c]}$, such that $q_{\ell,i,j}$ occurs in $\phi_{\ell,t}$ only if the $i$-th label in $\mathbf{y}$ is $j \in \mathcal{Y}$. We assume a canonical ordering over the variables occurring in each $\varphi_{\ell,t}$, for $t \in [R_\ell]$, and use $\varphi_{\ell,t,k}$ to refer to the $k$-th variable. We use $|\varphi_{\ell,t}|$ to denote the number of variables in $\varphi_{\ell,t}$.

Based on the above, finding a pseudo-label assignment for $(x_{\ell,1}, \ldots, x_{\ell,M})$ that adheres to $\sigma$ and $s_\ell$ reduces to finding an assignment to the variables in $\{q_{\ell,i,j}\}_{i \in [M], j \in [c]}$ that makes $\Phi_\ell$ hold. Previous work (Roth & Yih, 2007; Srikumar & Roth, 2023) has shown that we can cast satisfiability problems (as the one above) to linear programming problems. Therefore, instead of finding a Boolean true or false assignment to each $q_{\ell,i,j}$, we can find an assignment in $[0,1]$ for the real counterpart of $q_{\ell,i,j}$ denoted by $[q_{\ell,i,j}]$. Via associating the $[q_{\ell,i,j}]$'s to the entries in the $\mathbf{Q}_i$'s, i.e., $Q_i[\ell, j] = [q_{\ell,i,j}]$, we can solve the following linear program to perform pseudo-labeling:

$$\textbf{objective} \quad \min_{(\mathbf{Q}_1, \ldots, \mathbf{Q}_M)} \sum_{i=1}^M \langle -\log(\mathbf{P}_i), \mathbf{Q}_i \rangle,$$

$$\textbf{s.t.} \quad
\begin{aligned}
\sum_{t=1}^{R_\ell} [\alpha_{\ell,t}] &\geq 1, & \ell \in [n] \\
-|\varphi_{\ell,t}|[\alpha_{\ell,t}] + \sum_{k=1}^{|\varphi_{\ell,t}|} [\varphi_{\ell,t,k}] &\geq 0, & \ell \in [n], t \in [R_\ell] \\
-\sum_{k=1}^{|\varphi_{\ell,t}|} [\varphi_{\ell,t,k}] + [\alpha_{\ell,t}] &\geq (1 - |\varphi_{\ell,t}|), & \ell \in [n], t \in [R_\ell] \\
\sum_{j=1}^{c} [q_{\ell,i,j}] &= 1, & \ell \in [n], i \in [M] \\
[q_{\ell,i,j}] &\in [0,1], & \ell \in [n], i \in [M], j \in [c] \\
|\mathbf{Q}_i \cdot \mathbf{1}_n - n\widehat{\mathbf{r}}| &\leq \epsilon, & i \in [M]
\end{aligned}
\tag{6}$$

The objective in (6) aligns with our aim to find pseudo-labels close to the classifier's scores. The independence among the classifier's scores for different $\mathbf{x}_{\ell,i}$'s– recall that a classifier makes a prediction for each $\mathbf{x}_{\ell,i}$ independently of the other instances– justifies the sum over different $i$'s in the minimization objective. The first three constraints force the pseudo-labels for the $\ell$-th training sample to adhere to $\sigma$ and $s_\ell$, where the $\alpha_{\ell,t}$'s are Boolean variables introduced due to converting the $\Phi_\ell$'s into *conjunctive normal form* (CNF) using the Tseytin transformation (Tseitin, 1983). The fourth and the fifth constraint wants the pseudo-labels for each instance $x_{\ell,i}$ to sum up to one and lie in $[0, 1]$. Finally, the last constraint wants for each $i \in [M]$, the probability of predicting the $j$-th pseudo-label for an element in $\{x_{\ell,i}\}_{\ell \in [n]}$ to match the ratio estimates at hand $\widehat{r}_j$ up to some $\epsilon \geq 0$: the smaller $\epsilon$ gets, the stricter the adherence to $\widehat{\mathbf{r}}$ becomes. The detailed derivation of (6) is in Appendix D, as well as an example program formulation based on Example 1.1. Finally, Table 8 summarizes the notation.

To summarize, training-time mitigation works as follows: for each epoch, we split the training samples in $\mathcal{T}_\mathsf{P}$ into batches. For each batch $\{(x_{\ell,1}, \ldots, x_{\ell,M}, s_\ell)\}_{\ell \in [n]}$, we form matrices $\mathbf{P}_1, \ldots, \mathbf{P}_M$ by applying $f$ on the $x_{\ell,i}$'s and solve (6) to get the pseudo-label matrices $\mathbf{Q}_1, \ldots, \mathbf{Q}_M$. Finally, we train $f$ by minimizing the cross-entropy loss between $\mathbf{Q}_1, \ldots, \mathbf{Q}_M$ and $\mathbf{P}_1, \ldots, \mathbf{P}_M$. We will use LP to denote the above training technique.

**Remarks**. Our formulation in (6) is oblivious to $\widehat{\mathbf{r}}$, which can be estimated using either Algorithm 1 or any other technique, such as the moving average one from (Wang et al., 2022). Furthermore, the formulation in (6) allows us to find either hard or soft pseudo-labels: we can treat (6) as an integer linear program via forcing $[q_{\ell,i,j}]$ to lie in $\{0, 1\}$, instead of $[0, 1]$.

## 4.3 CAROT: TESTING-TIME IMBALANCE MITIGATION

We conclude this section with CAROT, an algorithm that mitigates learning imbalances at testing-time by modifying the model's scores to adhere to the estimated ratios $\widehat{\mathbf{r}}$. Incorporating $\widehat{\mathbf{r}}$ into the model's scores involves the design choices (i) and (ii) presented at the beginning of Section 4.2– challenges (iii) and (iv) are specific to training. Regarding (i), most existing testing-time mitigation algorithms algorithms (e.g., (Menon et al., 2021)) modify a model's scores at the level of individual instances. Regarding (ii), as we explained in Section 4.2, strictly enforcing $\widehat{\mathbf{r}}$ could also be problematic, as now, $\widehat{\mathbf{r}}$ may be also different from the label marginal underlying the test data.

Similarly to Section 4.2, we propose to adjust the model's scores for a whole batch of $n > 1$ test samples (represented by a matrix $\mathbf{P} \in \mathbb{R}^{n \times c}$) so that the adjusted scores $\mathbf{P}'$ roughly adhere to $\widehat{\mathbf{r}}$. Precisely, we propose to find $\mathbf{P}'$ that optimizes the following objective:

$$\min_{\mathbf{P}' \in \mathbb{R}_+^{n \times c}, \mathbf{P}'\mathbf{1}_c = \mathbf{1}_n} \langle -\log(\mathbf{P}), \mathbf{P}' \rangle + \tau \, \mathrm{KL}(\mathbf{P}'^\mathsf{T} \mathbf{1}_n \parallel n\widehat{\mathbf{r}}) - \eta H(\mathbf{P}') \tag{7}$$

The first term in (7) encourages $\mathbf{P}'$ to be close to the original scores. The second term encourages the column sums of $\mathbf{P}'$ to match $\widehat{\mathbf{r}}$, with $\tau > 0$ controlling adherence, where KL is the Kullback-Leibler divergence. This formulation leads to a *robust semi-constrained optimal transport* (RSOT) problem (Le et al., 2021). The regularizer $\eta H(\mathbf{P}')$, where $H$ denotes entropy, allows to approximate the optimal solution using the robust semi-Sinkhorn algorithm (Le et al., 2021), leading to CAROT (*Confidence-Adjustment via Robust semi-constrained Optimal Transport*), see Algorithm 2.

In Algorithm 2, $B(\mathbf{u}, \mathbf{v})$ denotes an $n \times c$ matrix whose $(i, j)$ cell is computed as a function of $\mathbf{u}$ and $\mathbf{v}$ by $\exp(u_i + v_j + \log(P_{ij})/\eta)$. In each iteration, the algorithm alternates between updating the $c$-dimensional vector $\mathbf{v}$ and the $n$-dimensional vector $\mathbf{u}$. The former update, which is computed as $\mathbf{v} \leftarrow \frac{\eta\tau}{\eta+\tau}\left(\frac{\mathbf{v}}{\eta} + \log(n\widehat{\mathbf{r}}) - \log(\mathbf{b})\right)$, forces $B(\mathbf{u}, \mathbf{v})$ to adhere to $\widehat{\mathbf{r}}$; the latter, which is computed as $\mathbf{u} \leftarrow \eta\left(\frac{\mathbf{u}}{\eta} + \log(\mathbf{1}_n) - \log(\mathbf{a})\right)$, forces the elements in each row of $B(\mathbf{u}, \mathbf{v})$ to add to one. Matrix $B(\mathbf{u}, \mathbf{v})$ converges to the optimal solution to (7) when $N_{\text{iter}}$ goes to infinity (Le et al., 2021).

**Choice of $\eta$ and $\tau$.** In practice, we use a small *partially labeled* validation set to choose $\eta$ and $\tau$. Doing so, the validation set can be obtained by splitting the training set of partially labelled data $\mathcal{T}_\mathsf{P}$.

**Guarantees.** CAROT minimizes (7) under a polynomial number of iterations, see (Le et al., 2021). Being a testing-time technique, this is the only guarantee that CAROT can reasonably provide.

## 5 EXPERIMENTS

**Baselines.** We focus on scenarios from neurosymbolic learning due to the increasing interest on the topic. We consider the state-of-the-art loss *semantic loss* (SL) (Xu et al., 2018; Wang et al., 2023b; Huang et al., 2021) for MI-PLL training and use the engine Scallop that performs MI-PLL training using that loss (Huang et al., 2021). Since there are no prior MI-PLL techniques for mitigating imbalances at testing-time, we consider Logit Adjustment (LA) (Menon et al., 2021) as a competitor to CAROT. The notation +A, for an algorithm A $\in$ {LA, CAROT}, means that the scores of a baseline model are modified at testing-time via A. We do not assume access to a validation set of gold labelled data, applying LA and CAROT using the estimate $\widehat{\mathbf{r}}$ obtained via Algorithm 1. However, we use a validation set of partially labelled data to run Algorithm 1. We also carry experiments with RECORDS (Hong et al., 2023), a technique that mitigates imbalances at training-time for standard PLL (no previous MI-PLL training-time baseline exists). We use SL+RECORDS when a classifier has been trained using RECORDS in conjunction with SL. RECORDS acts as a competitor to LP. Notice that the imbalance mitigation technique from (Wang et al., 2022), SOLAR, cannot act as a competitor to our proposed techniques (see Appendix E for a detailed discussion on SOLAR). Finally, we carry experiments using LP, see Section 4.2. We use LP(ALG1) and LP(EMP), when LP is applied using the ratios obtained via Algorithm 1 and via the approximation from (Wang et al., 2022).

**Benchmarks.** We carry experiments using an MI-PLL benchmark previously used in the neurosymbolic literature (Manhaeve et al., 2018; 2021b; Huang et al., 2021; Li et al., 2023a), namely MAX-$M$, as well as a newly introduced, called Smallest Parent. Training samples in MAX-$M$ are as described in Example 1.1. We vary $M$ to $\{3, 4, 5\}$ and use the MNIST benchmark to obtain training and testing instances. In Smallest Parent, training samples are of the form $(x_1, x_2, p)$, where $x_1$ and $x_2$ are CIFAR-10 images and $p$ is the most immediate common ancestor of $y_1$ and $y_2$, assuming the classes form a hierarchy. To simulate long-tail phenomena (denoted as **LT**), we vary the imbalance ratio $\rho$ of the distributions of the input instances as in (Cao et al., 2019; Wang et al., 2022): $\rho = 0$ means that the hidden label distribution is unmodified and balanced. Despite looking simply at a first glance, our scenarios are quite challenging. First, the pre-image of $\sigma$ may be particularly large, making the supervision rather weak, e.g., in the MAX-5 scenario, there are $5 \times 9^4$ candidate label vectors when the partial label is 9. Second, the transition functions may exacerbate the imbalances in the hidden labels, with the probability of certain partial labels getting very close to zero. For instance, in the MAX-5 scenario, the probability of the partial label zero is $10^{-5}$ when $\rho = 0$. This probability becomes even smaller when $\rho = 50$. Each cell shows mean accuracy and standard deviation over three different runs. The results of our analysis are summarized in Table 1, Table 2 and Figure 3. Results on more neurosymbolic scenarios and a further analysis are in the appendix.

Table 1: Experimental results for MAX-$M$ using $m_\mathsf{P} = 3000$.

| Algorithms | Original $\rho = 0$ | | | LT $\rho = 15$ | | | LT $\rho = 50$ | | |
| | $M = 3$ | $M = 4$ | $M = 5$ | $M = 3$ | $M = 4$ | $M = 5$ | $M = 3$ | $M = 4$ | $M = 5$ |
|---|---|---|---|---|---|---|---|---|---|
| SL | $84.15 \pm 11.92$ | $73.82 \pm 2.36$ | $59.88 \pm 5.58$ | $71.25 \pm 4.48$ | $66.98 \pm 3.2$ | $55.06 \pm 5.21$ | $66.74 \pm 5.42$ | $67.71 \pm 11.58$ | $55.74 \pm 2.58$ |
| + LA | $84.17 \pm 11.95$ | $73.82 \pm 2.36$ | $59.88 \pm 5.58$ | $70.80 \pm 4.52$ | $66.98 \pm 3.20$ | $54.53 \pm 5.74$ | $66.57 \pm 5.09$ | $61.10 \pm 3.95$ | $52.47 \pm 8.06$ |
| + CAROT | $84.57 \pm 11.50$ | $73.08 \pm 3.10$ | $60.26 \pm 5.20$ | $74.95 \pm 3.45$ | $67.44 \pm 2.74$ | $55.80 \pm 4.47$ | $68.16 \pm 4.00$ | $68.25 \pm 6.14$ | $57.29 \pm 14.17$ |
| RECORDS | $85.56 \pm 7.25$ | $75.11 \pm 0.77$ | $59.43 \pm 6.61$ | $55.47 \pm 20.45$ | $53.34 \pm 16.66$ | $52.40 \pm 7.95$ | $70.20 \pm 7.65$ | $66.05 \pm 13.90$ | $59.93 \pm 4.86$ |
| + LA | $87.63 \pm 5.11$ | $75.11 \pm 0.77$ | $59.28 \pm 6.76$ | $54.90 \pm 20.16$ | $54.46 \pm 15.54$ | $51.25 \pm 9.09$ | $70.09 \pm 7.26$ | $65.78 \pm 14.18$ | $59.93 \pm 4.86$ |
| + CAROT | $90.97 \pm 2.03$ | $75.94 \pm 0.91$ | $60.45 \pm 7.78$ | $54.32 \pm 21.85$ | $62.74 \pm 8.14$ | $55.85 \pm 4.61$ | $71.46 \pm 6.4$ | $71.25 \pm 8.70$ | $63.64 \pm 5.92$ |
| LP(EMP) | $94.97 \pm 1.32$ | $77.86 \pm 4.22$ | $55.27 \pm 11.27$ | $75.83 \pm 5.26$ | $69.67 \pm 5.47$ | $59.25 \pm 7.27$ | $77.16 \pm 3.46$ | $70.06 \pm 10.73$ | $56.79 \pm 1.58$ |
| + LA | $94.69 \pm 1.60$ | $77.91 \pm 4.16$ | $55.34 \pm 11.19$ | $75.77 \pm 5.32$ | $68.92 \pm 3.96$ | $58.49 \pm 5.74$ | $77.1 \pm 3.52$ | $69.76 \pm 10.31$ | $56.81 \pm 1.56$ |
| + CAROT | $95.07 \pm 1.20$ | $75.53 \pm 7.42$ | $53.07 \pm 12.99$ | $76.38 \pm 4.72$ | $69.74 \pm 5.51$ | $59.56 \pm 8.14$ | $77.58 \pm 3.04$ | $70.11 \pm 10.34$ | $57.09 \pm 1.90$ |
| LP(ALG1) | $96.09 \pm 0.41$ | $78.34 \pm 4.80$ | $59.91 \pm 6.63$ | $74.51 \pm 9.13$ | $69.14 \pm 1.82$ | $56.81 \pm 3.74$ | $72.23 \pm 11.49$ | $69.28 \pm 11.78$ | $63.67 \pm 7.04$ |
| + LA | $95.81 \pm 0.74$ | $78.97 \pm 4.09$ | $59.98 \pm 6.56$ | $74.26 \pm 9.06$ | $68.73 \pm 2.23$ | $56.37 \pm 3.13$ | $72.23 \pm 11.49$ | $69.21 \pm 11.86$ | $63.67 \pm 7.04$ |
| + CAROT | $96.13 \pm 0.38$ | $80.78 \pm 2.36$ | $59.71 \pm 6.35$ | $77.05 \pm 7.00$ | $69.19 \pm 1.81$ | $59.76 \pm 7.24$ | $74.82 \pm 10.18$ | $74.30 \pm 7.54$ | $64.39 \pm 6.43$ |

Table 2: Experimental results for Smallest Parent using $m_\mathsf{P} = 10000$.

| Algorithms | Original $\rho = 0$ | LT $\rho = 5$ | LT $\rho = 15$ | LT $\rho = 50$ | Algorithms | Original $\rho = 0$ | LT $\rho = 5$ | LT $\rho = 15$ | LT $\rho = 50$ |
|---|---|---|---|---|---|---|---|---|---|
| SL | $69.82 \pm 0.53$ | $67.94 \pm 0.40$ | $69.04 \pm 0.03$ | $74.65 \pm 0.44$ | LP(EMP) | $79.41 \pm 1.33$ | $79.24 \pm 1.03$ | $68.40 \pm 1.90$ | $70.29 \pm 1.62$ |
| + LA | $69.83 \pm 0.53$ | $67.93 \pm 0.41$ | $68.70 \pm 0.30$ | $74.62 \pm 0.36$ | + LA | $79.41 \pm 1.33$ | $79.24 \pm 1.03$ | $68.40 \pm 1.90$ | $70.29 \pm 1.62$ |
| + CAROT | $69.82 \pm 0.53$ | $67.93 \pm 0.41$ | $68.70 \pm 0.41$ | $74.15 \pm 0.47$ | + CAROT | $79.41 \pm 1.33$ | $79.28 \pm 0.91$ | $77.10 \pm 1.74$ | $80.71 \pm 1.50$ |
| RECORDS | $48.71 \pm 3.90$ | $48.15 \pm 4.56$ | $50.14 \pm 1.10$ | $55.12 \pm 1.40$ | LP(ALG1) | $80.23 \pm 0.70$ | $81.27 \pm 0.71$ | $81.99 \pm 0.51$ | $83.44 \pm 0.48$ |
| + LA | $54.12 \pm 2.00$ | $45.48 \pm 2.31$ | $56.83 \pm 1.30$ | $60.87 \pm 1.20$ | + LA | $80.20 \pm 0.74$ | $81.26 \pm 0.72$ | $81.99 \pm 0.51$ | $83.44 \pm 0.48$ |
| + CAROT | $68.16 \pm 0.47$ | $69.04 \pm 0.74$ | $71.70 \pm 0.84$ | $75.69 \pm 0.90$ | + CAROT | $68.90 \pm 11.09$ | $76.38 \pm 5.68$ | $82.00 \pm 0.51$ | $83.44 \pm 0.48$ |

**Conclusions.** We observed many interesting phenomena: (i) training-time mitigation can significantly improve the accuracy; (ii) state-of-the-art on training-time mitigation might not be appropriate for MI-PLL; (iii) approximate techniques for estimating **r** can sometimes be more effective when used for training-time mitigation; (iv) testing-time mitigation can substantially improve the accuracy of a

classifier; however, it tends to be less effective than training-time mitigation; (v) CAROT may be sensitive to the quality of estimated ratios $\hat{\mathbf{r}}$; (vi) Algorithm 1 offers quite accurate marginal estimates.

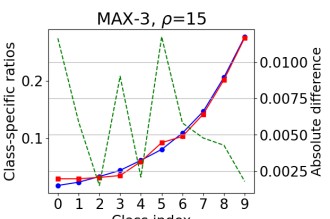 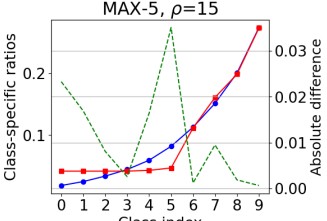 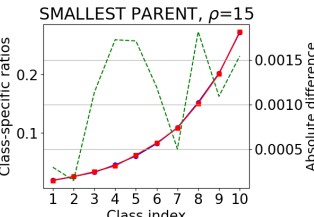

Figure 3: Accuracy of the marginal estimates computed by Algorithm 1. Blue denotes the gold ratios, red the estimated ones, and green the absolute difference between the gold and estimated ratios.

Starting from the last conclusion, Figure 3 shows that Algorithm 1 offers quite accurate estimates even in challenging scenarios with high imbalance ratios. Regarding (i), let us focus on Table 2. We can see that both LP(EMP) and LP(ALG1) lead to higher accuracy than models trained exclusively via SL. For example, when $\rho = 5$ in Smallest Parent, the mean accuracy obtained via training under SL is 67.94%; the mean accuracy increases to 79.24% under LP(EMP) and to 81.27% under LP(ALG1). In MAX-4, the mean accuracy under SL is 55.48%, increasing to 78.56% under LP(ALG1). Regarding (ii), consider again Table 2: when RECORDS is applied jointly with SL, the accuracy of the model can substantially drop, e.g., when $\rho = 5$ in Table 2, the mean accuracy drops from 67.94% to 48.15%. In the MAX-$M$ scenarios, RECORDS seems to improve over SL; however, for certain scenarios the accuracy drops drastically (e.g., for $\rho = 15$). The above stresses the importance of LP(Section 4.2).

Let's move to (iii). In most of the cases, LP(ALG1) leads to higher accuracy than LP(EMP). However, the opposite may also hold in some cases. One such example is MAX-3 for $\rho = 50$: the mean accuracy for the baseline model is 66.74%, increasing to 72.23% under LP(ALG1) and to 77.16% under LP(EMP). A similar phenomenon is observed for $\rho = 15$ for the same scenario. The above suggests that there can be cases where employing the gold ratios (Algorithm 1 produces estimates that converge to the gold ratios, see Proposition C.1) may not always be the best solution. A similar observation is made by the authors of RECORDS (Hong et al., 2023). One cause of this phenomenon is the high number of classification errors during the initial stages of learning. Those classification errors can become higher in

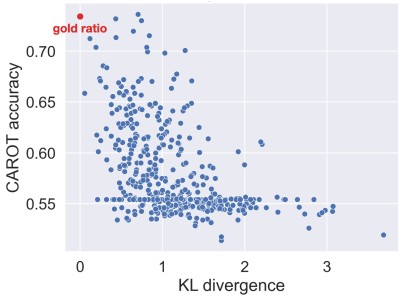

Figure 4: Impact of the label ratio quality on CAROT's performance.

our experimental setting, as in MAX-$M$, we only consider a subset of the pre-images of each partial label to compute SL and (6), to reduce the computational overhead of computing all pre-images.

We conclude with CAROT. Tables 1 and 2 show that CAROT can be more effective than LA. For example, in the MAX-3 scenarios and $\rho = 50$, the mean accuracy is 66.74% under SL, drops to 66.57% under SL+LA and increases to 68.16% under SL+CAROT. In Smallest Parent and $\rho = 50$, the mean accuracy of LP(EMP) increases from 70.29% to 80.71% under CAROT; LA has no impact. CAROT also improves the accuracy of RECORDS models, often, by a large margin. For example, for Smallest Parent and $\rho = 15$, the mean accuracy of a RECORDS-based trained model increases from 50.14% to 71.70% when CAROT is applied. CAROT is also consistently better than LA when applied on top of RECORDS. However, there can be cases where both LA and CAROT drop the accuracy of the baseline model. One such example is met in Smallest Parent and $\rho = 5$: the mean accuracy under LP(ALG1) is 81.27% and drops to 76.38% when CAROT is applied.

We analyse the sensitivity of CAROT under the quality of the input $\hat{\mathbf{r}}$, where quality is measured by means of the KL divergence to $\mathbf{r}$. Figure 4 shows the accuracy of an MNIST model (trained with the MAX-3 dataset), when CAROT is applied at testing-time using 500 randomly generated ratios $\hat{\mathbf{r}}$ of varying quality. We observe that CAROT's effectiveness drops as the estimated marginal diverges more from $\mathbf{r}$. Also, the performance can decrease by more than 10% with only a small perturbation in the KL divergence. This instability may be the reason CAROT fails to improve a base model.

**Training- vs testing-time mitigation.** CAROT is a more lightweight technique, relying on the polynomial complexity, semi-Sinkhorn algorithm (Le et al., 2021). However, as the empirical results suggest, CAROT may lead to lower classification accuracy in comparison to LP. On the contrary, LP *may* increase the training overhead over the state-of-the-art– that is applying the top-$k$ SL per training sample (Xu et al., 2018; Wang et al., 2023b). This is because when $k$ is fixed, the complexity to compute the SL is polynomial; in contrast, solving (6), which is a linear program calculated out of a batch of samples, is an NP-hard problem. When the SL runs *without* approximations though and the pre-image of $\sigma$ is very large, the complexity of SL is worst case #P-complete per training sample (Chavira & Darwiche, 2008), making (6) a more computationally efficient approach.

## 6 RELATED WORK

An extended version and more detailed comparison against the related work is in Appendix E.

**Long-tail supervised learning**. Two supervised learning techniques related to our work are LA (Menon et al., 2021) and OTLM (Peng et al., 2022). Both aim at testing-time mitigation. LA modifies the classifier's scores by subtracting the gold ratios. CAROT can be substantially more effective than LA, see Section 5. OTLM assumes that the marginal $\mathbf{r}$ is known, resorting to an OT formulation for adjusting the classifier's scores. In contrast, we propose a statistically consistent technique to estimate $\mathbf{r}$, see Section 4.1, and resort to RSOT to accommodate for noisy $\widehat{\mathbf{r}}$'s.

**Long-tail PLL**. The authors in (Cour et al., 2011) showed that certain classes are harder to learn than others in standard PLL. We are the first to extend those results under MI-PLL. The only two works in the intersection of long-tail learning and (single-instance) PLL are RECORDS (Hong et al., 2023) and SOLAR (Wang et al., 2022). RECORDS modifies the classifier's scores using the same idea with LA. It employs a momentum-updated prototype feature to estimate $\widehat{\mathbf{r}}$. Unlike LP, RECORDS does not take into account the constraints coming from MI-PLL. Section 5 shows that RECORDS is less effective than our proposals, degrading the baseline accuracy on multiple occasions. SOLAR relies on standard OT to assign pseudo-labels to instances, in contrast to our formulation in (6). Also, SOLAR uses an averaging technique to estimate $\mathbf{r}$, as opposed to Algorithm 1.

**MI-PLL**. We close with some recent theoretical results on MI-PLL. The authors in (Marconato et al., 2023; 2024) characterize *reasoning shortcuts* in MI-PLL. In contrast, our work provides class-specific error bounds, formally characterizing learning imbalances in MI-PLL. It is worth noting that the authors in (Tang et al., 2024a;b) use the term multi-instance partial-label learning to describe their learning setting. The differences with ours (see Section 2) are as follows. First, the objective in (Tang et al., 2024a;b) is to learn a *bag classifier*, i.e., a classifier $f : 2^{\mathcal{X}} \rightarrow \mathcal{Y}$, and not an instance classifier. Second, unlike our setting, in (Tang et al., 2024a;b), the training samples are of the form $(\mathbf{X}, \mathbf{S})$, where $\mathbf{X}$ is a *bag* of instances and $\mathbf{S}$ is a *bag* of labels for the *whole* $\mathbf{X}$. Due to the above differences, the formulation in (Tang et al., 2024a;b) cannot capture the neurosymbolic learning setting in (Manhaeve et al., 2018; Dai et al., 2019; Tsamoura et al., 2021; Li et al., 2023a).

## 7 CONCLUSIONS AND FUTURE WORK

**Comments on the theory.** Our analysis in Section 3 assumes that the probability of misclassifying an instance $x$ only dependents on its class. This assumption is also adopted in other learning settings, such as *noisy label learning* (Zhang et al., 2021; Patrini et al., 2017). Although there are more complex scenarios where this assumption does not hold, our theory stands as an over-approximation to those scenarios, similarly to the connection between class- and instance-dependent noisy label learning. Furthermore, our formulation in (3) can be extended to cases where the correlations among the instances $(x_1, \ldots, x_M)$ of each training sample are *weak*, i.e., have very few correlations. Extending our analysis in the general non-i.i.d. setting is an important direction for future research.

Our work is the first to theoretically characterize and mitigate learning imbalances in MI-PLL. Our theoretical characterization complements the existing theory in long-tail learning, identifying and addressing the unique challenges in MI-PLL. Additionally, we contributed an LP-based and an RSOT-based mitigation technique that both outperform state-of-the-art in long-tail learning. Our empirical analysis unveiled two topics for future research: *computing marginal for testing-time mitigation* and *designing more effective testing-time mitigation techniques*. Another important future direction is to look into scalability, as for scenarios with a large number of classes, it may be computationally expensive to run Algorithm 1 or Algorithm 2.

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

## APPENDIX ORGANIZATION

Our appendix is organized as follows:

## A    EXTENDED PRELIMINARIES

**Optimal transport.** Let $Z_1$ and $Z_2$ be two discrete random variables over $[m_1]$ and $[m_2]$. For $i \in [2]$, vector $\mathbf{b}^i \in \mathbb{R}_+^{m_i}$ denotes the probability distribution of $Z_i$, i.e., $\mathbb{P}(Z_i = m_j) = b_j^i$, for each $j \in [m_i]$. Let $U$ be the set of matrices defined as $\{\mathbf{Q} \in \mathbb{R}_+^{m_1 \times m_2} | \mathbf{Q}\mathbf{1}_{m_1} = \mathbf{b}^2, \mathbf{Q}\mathbf{1}_{m_2} = \mathbf{b}^1\}$. The *optimal transport* (OT) problem (Peyré & Cuturi, 2020) asks us to find the matrix $\mathbf{Q} \in U$ that maximizes a linear object subject to marginal constraints, namely

$$\min_{\mathbf{Q} \in U} \langle \mathbf{P}, \mathbf{Q} \rangle \tag{8}$$

Assume that we are strict in enforcing the probability distribution $\mathbf{b}^1$, but not in enforcing $\mathbf{b}^2$. The *robust semi-constrained optimal transport* (RSOT) problem (Le et al., 2021) aims to find:

$$\min_{\mathbf{Q} \in U'} \langle \mathbf{P}, \mathbf{Q} \rangle + \tau \mathrm{KL}(\mathbf{Q}\mathbf{1}_{m_1} || \mathbf{b}^2) \tag{9}$$

where $U' = \{\mathbf{Q} \in \mathbb{R}_+^{m_1 \times m_2} | \mathbf{Q}\mathbf{1}_{m_2} = \mathbf{b}^1\}$ and $\tau > 0$ is a regularization parameter. The solution to (9) can be approximated in polynomial time using the *robust semi-Sinkhorn algorithm* from (Le et al., 2021), which generalizes the classical Sinkhorn algorithm (Cuturi, 2013) for OT.

## B    PROOFS AND DETAILS FOR SECTION 3

### B.1    PROOFS

**Proposition 3.1** (Class-specific risk bound). *For any $j \in \mathcal{Y}$, we have that $R_j(f) \leq \Phi_{\sigma,j}(R_\mathsf{P}(f; \sigma))$.*

*Proof.* This result directly follows from the definition of the program (3). $\qquad\square$

**Proposition 3.3.** *Let $d_{[\mathcal{F}]}$ be the Natarajan dimension of $[\mathcal{F}]$. Given a confidence level $\delta \in (0, 1)$, we have that $R_j(f) \leq \Phi_{\sigma,j}(\widetilde{R}_\mathsf{P}(f; \sigma, \mathcal{T}_\mathsf{P}, \delta))$ with probability $1 - \delta$ for any $j \in [c]$, where*

$$\widetilde{R}_\mathsf{P}(f; \sigma, \mathcal{T}_\mathsf{P}, \delta) = \widehat{R}_\mathsf{P}(f; \sigma, \mathcal{T}_\mathsf{P}) + \sqrt{\frac{2 \log(e m_\mathsf{P}/2 d_{[\mathcal{F}]} \log(6Mc^2 d_{[\mathcal{F}]}/e))}{m_\mathsf{P}/2 d_{[\mathcal{F}]} \log(6Mc^2 d_{[\mathcal{F}]}/e)}} + \sqrt{\frac{\log(1/\delta)}{2m_\mathsf{P}}} \tag{4}$$

*Proof.* To start with, let $L_\sigma \circ [\mathcal{F}]$ be the function space that maps a (training) example $(\mathbf{x}, s)$ to its partial loss defined as follows:

$$L_\sigma \circ [\mathcal{F}] := \{(\mathbf{x}, s) \mapsto L_\sigma([f](\mathbf{x}), s) | f \in \mathcal{F}\} \tag{10}$$

The standard generalization bound with VC dimension (see, for example, Corollary 3.19 of (Mohri et al., 2018)) implies that:

$$R_{\mathsf{P}}(f) \leq \widehat{R}_{\mathsf{P}}(f; \mathcal{T}_{\mathsf{P}}) + \sqrt{\frac{2\log(em_{\mathsf{P}}/d_{\mathrm{VC}}(L_\sigma \circ [\mathcal{F}]))}{m_{\mathsf{P}}/d_{\mathrm{VC}}(L_\sigma \circ [\mathcal{F}])}} + \sqrt{\frac{\log(1/\delta)}{2m_{\mathsf{P}}}} \tag{11}$$

where $d_{\mathrm{VC}}(\cdot)$ is the VC dimension. For simplicity, let $d = d_{\mathrm{VC}}(L_\sigma \circ [\mathcal{F}])$ and $d_{[\mathcal{F}]}$ be the Natarajan dimension of $[\mathcal{F}]$. Using a similar argument as in (Wang et al., 2023b), given any $d$ samples in $\mathcal{X}^M \times \mathcal{O}$ using $[\mathcal{F}]$, we let $N$ be the maximum number of distinct ways to assign label vectors (in $\mathcal{Y}^M$) to these $d$ samples. Then, the definition of VC-dimension implies that:

$$2^d \leq N \tag{12}$$

On the other hand, these $d$ samples contain $Md$ input instances in $\mathcal{X}$. By Natarajan's lemma (see, for example, Lemma 29.4 of (Shalev-Shwartz & Ben-David, 2014)), we have that:

$$N \leq (Md)^{d_{[\mathcal{F}]}} c^{2d_{[\mathcal{F}]}} \tag{13}$$

Combining (13) with the above equations, it follows that

$$(Md)^{d_{[\mathcal{F}]}} c^{2d_{[\mathcal{F}]}} \geq N \geq 2^d \tag{14}$$

Taking the logarithm on both sides, we have that:

$$d_{[\mathcal{F}]} \log(Md) + 2d_{[\mathcal{F}]} \log c \geq d \log 2 \tag{15}$$

Taking the first-order Taylor series expansion of the logarithm function at the point $6d_{[\mathcal{F}]}$, we have:

$$\log(d) \leq \frac{d}{6d_{[\mathcal{F}]}} + \log(6d_{[\mathcal{F}]}) - 1 \tag{16}$$

Therefore,

$$
\begin{aligned}
d \log 2 &\leq d_{[\mathcal{F}]} \log d + d_{[\mathcal{F}]} \log M + 2d_{[\mathcal{F}]} \log c \\
&\leq d_{[\mathcal{F}]} \left( \frac{d}{6d_{[\mathcal{F}]}} + \log(6d_{[\mathcal{F}]}) - 1 \right) + d_{[\mathcal{F}]} \log M + 2d_{[\mathcal{F}]} \log c \\
&= \frac{d}{6} + d_{[\mathcal{F}]} \log(6Mc^2 d_{[\mathcal{F}]}/e)
\end{aligned}
\tag{17}
$$

Rearranging the inequality yields

$$
\begin{aligned}
d &\leq \frac{d_{[\mathcal{F}]} \log(6Mc^2 d_{[\mathcal{F}]}/e)}{\log 2 - 1/6} \\
&\leq 2d_{[\mathcal{F}]} \log(6Mc^2 d_{[\mathcal{F}]}/e)
\end{aligned}
\tag{18}
$$

as claimed. □

**Proposition 3.5.** *If $\sigma$ is $M$-unambiguous, then the risk of $f$ can be bounded by*

$$R(f) \leq \sqrt{\mathbf{w}^\mathsf{T}(D(\boldsymbol{\Sigma}_{\sigma,\mathbf{r}}))^\dagger \mathbf{w} R_{\mathsf{P}}(f; \sigma)} = \sqrt{c(c-1)R_{\mathsf{P}}(f; \sigma)} \tag{5}$$

*which coincides with Lemma 1 from (Wang et al., 2023b) for $M = 2$, where $\mathbf{w} := \sum_{j=1}^c r_j \mathbf{w}_j$.*

*Proof.* Since $\mathbf{w} := \sum_{i=1}^c r_i \mathbf{w}_i$, we have $R(f) = \mathbf{w}^\mathsf{T}\mathbf{h}$. Then, we consider the following relaxed program:

$$
\begin{aligned}
\max_{\mathbf{h}} \quad & \mathbf{w}^\mathsf{T}\mathbf{h} \\
\text{s.t.} \quad & \mathbf{h}^\mathsf{T} D(\boldsymbol{\Sigma}_{\sigma,\mathbf{r}})\mathbf{h} \leq R_{\mathsf{P}}
\end{aligned}
\tag{19}
$$

where $D(\boldsymbol{\Sigma}_{\sigma,\mathbf{r}})$ is the diagonal part of $\boldsymbol{\Sigma}_{\sigma,\mathbf{r}}$, namely:

$$D(\boldsymbol{\Sigma}_{\sigma,\mathbf{r}}) = [r_i r_j \mathbb{1}\{i = j\} \mathbb{1}\{i \not\equiv j \pmod{c}\}]_{i \in [c^2], j \in [c^2]} \tag{20}$$

In other words, $D(\boldsymbol{\Sigma}_{\sigma,\mathbf{r}})$ encodes all the partial risks that is caused by repeating the same type of misclassification twice. On the other hand, the $M$-unambiguity condition ensures that each type of

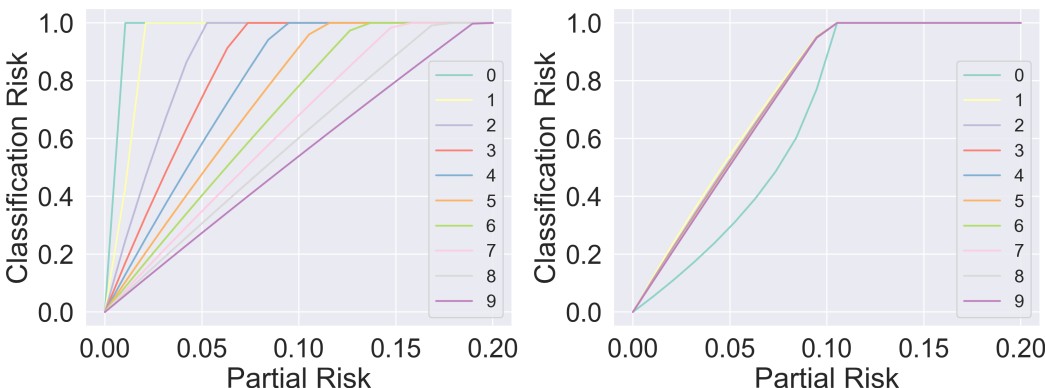

Figure 5: Class-specific upper bounds obtained via (3). (left) $\mathcal{D}_Y$ is uniform. (right) $\mathcal{D}_{P_S}$ is uniform. (Enlarged version of Figure 2).

misclassification, when repeated twice, leads to a misclassification of the partial label. Therefore, $\mathbf{w} \in \text{Range}(D(\mathbf{\Sigma}_{\sigma,\mathbf{r}}))$.

Problem (19) is a special case of the single constraint quadratic optimization problem. Then, the fact that $\mathbf{w} \in \text{Range}(D(\mathbf{\Sigma}_{\sigma,\mathbf{r}}))$ implies that the dual function of this problem (with dual variable $\lambda$) is

$$g(\lambda) = \lambda R_{\mathsf{P}} + \frac{\mathbf{w}^{\mathsf{T}}(D(\mathbf{\Sigma}_{\sigma,\mathbf{r}}))^{\dagger}\mathbf{w}}{4\lambda} \tag{21}$$

where $(D(\mathbf{\Sigma}_{\sigma,\mathbf{r}}))^{\dagger}$ is the pseudo-inverse, namely

$$(D(\mathbf{\Sigma}_{\sigma,\mathbf{r}}))^{\dagger} = [(r_i r_j)^{-1}\mathbb{1}\{i = j\}\mathbb{1}\{i \not\equiv j \;(\text{mod } c)\}]_{i \in [c^2], j \in [c^2]} \tag{22}$$

Therefore,

$$\mathbf{w}^{\mathsf{T}}(D(\mathbf{\Sigma}_{\sigma,\mathbf{r}}))^{\dagger}\mathbf{w} = c(c-1) \tag{23}$$

According to Appendix B of (Boyd & Vandenberghe, 2004), strong duality holds for this problem. Therefore, the optimal value is given exactly as

$$\inf_{\lambda \geq 0} g(\lambda) = 2\sqrt{\frac{c(c-1)}{4}R_{\mathsf{P}}} = \sqrt{c(c-1)R_{\mathsf{P}}} \tag{24}$$

as claimed. $\qquad\qquad\qquad\qquad\qquad\qquad\qquad\qquad\qquad\qquad\qquad\qquad\qquad\qquad\qquad\square$

### B.2  FURTHER DISCUSSION ON OUR BOUNDS

Intuitively, the difficulty of learning is affected by (i) the distribution of partial labels in $\mathbf{D}_{\mathsf{P}}$ and (ii) the size of the pre-image of $\sigma$ for each partial label. These two factor are reflected in our risk-specific bounds. Let us continue with the analysis in Example 3.2.

**Example B.1** (Cont' Example 3.2). *Let us start with* CASE 1. *In this case, our class-specific bounds suggest that learning the class zero is more difficult than learning class nine despite that both hidden labels $y_1$ and $y_2$ are uniform in $\{0, \ldots, 9\}$, see left side of Figure B.2. The root cause of this learning imbalance is $\sigma$ and its characteristics. In particular, the partial labels that result after independently drawing pairs of MNIST digits and applying $\sigma$ on their gold labels are long-tailed, with $s = 0$ occurring with probability $1/100$ and $s = 9$ occurring with probability $17/100$ in the training data. Hence, we have more supervision to learn class nine than to learn zero.*

*Now, let us move to* CASE 2. *In this case, our class-specific bounds suggest that learning class zero is the easiest to learn, see right side of Figure B.2. This is because of two reasons. First, the partial labels are uniform and hence, we have the same supervision to learn all classes. Second, the pre-image of $\sigma$ for different partial labels is very different. Regarding the second reason, partial label $s = 0$ provides much stronger supervision than partial label $s = 9$: when $s = 0$, we have direct supervision ($s = 0$ implies $y_1 = y_2 = 0$); in contrast, when $s = 9$ this only means that either $y_1 = 9$ and $y_2$ is any label in $\{0, \ldots, 9\}$, or vice versa.*

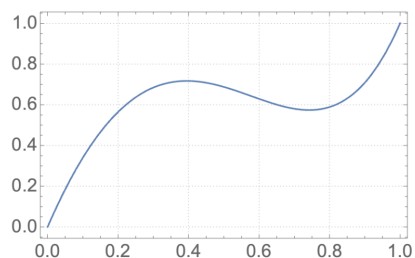

Figure 6: Plot of function $t \mapsto t^4 + 6t^2(1-t)^2 + 4t(1-t)^3$.

The above shows that $\sigma$ (i) can lead to imbalanced partial labels even if the hidden labels are uniformly distributed and (ii) may provide supervision signals of very different strengths. Hence, learning in MI-PLL is *inherently imbalanced* due to $\sigma$.

### B.3 Details on plotting Figure 2

In this subsection, we describe the steps we followed to create the plots in Figure 2. We generated the curves shown in each figure by plotting 20 evenly spaced points within the partial risk interval $R_\mathsf{P} \in [0, 0.2]$. To obtain the value of the classification risk at each point, we solved the optimization program (3) by using the COBYLA optimization algorithm implemented by the `scipy.optimize` package. To mitigate numerical instability, for each point, we ran the optimization solver ten times and dropped all the invalid results that were not in the range $[0, 1]$. The median of the remaining valid results was then taken as the solution to (3).

## C Further details on Algorithm 1

The estimate $\widehat{\mathbf{r}}$ given by Algorithm 1 can be viewed as a method to find the maximum likelihood estimation whose consistency is guaranteed under suitable conditions. The most critical one is the invertibility of $\Psi_\sigma$. The invertibility is satisfied by practical transitions as the one from Example 1.1, but may fail to hold for certain transitions even if the $M$-unambiguity condition (Wang et al., 2023b) holds. We will provide one such example later in this section.

Suppose that the backprobagation step in Algorithm 1 can effectively find the maximum likelihood estimator. For a real $\epsilon > 0$, let $\Delta_c^\epsilon$ be the shrinked probability simplex defined as $\Delta_c^\epsilon := \{\mathbf{r} \in \Delta_c | r_j \geq \epsilon \, \forall j \in [c]\}$. Let $\widehat{\mathbf{r}}^*_{m_\mathsf{P}} := \operatorname{argmin}_{\widehat{\mathbf{r}} \in \Delta_c^\epsilon} \sum_{j=1}^{c_S} \bar{p}_j \log[\Psi_\sigma(\widehat{\mathbf{r}})]_j$ be the maximum likelihood estimation. The following holds:

**Proposition C.1** (Consistency). *If there exists an $\epsilon > 0$, such that $\mathbf{r} \in \Delta_c^\epsilon$ and $\Psi_\sigma$ is injective in $\Delta_c^\epsilon$, then $\widehat{\mathbf{r}}^*_{m_\mathsf{P}} \to \mathbf{r}$ in probability as $m_\mathsf{P} \to \infty$.*

*Proof.* Let $\Delta_{c_S}^{\sigma,\epsilon} := \{\Psi_\sigma(\mathbf{r}) | \mathbf{r} \in \Delta_c^\epsilon\}$ be the image of $\Psi_\sigma$ on $\Delta_c^\epsilon$. The set $\Delta_{c_S}^{\sigma,\epsilon}$ is a compact subset in $\mathbb{R}^{c_S}$. For any partial label $a_j \in \mathcal{S}$, let $H(a_j, \mathbf{r}) := -\log([\Psi_\sigma(\mathbf{r})]_j)$ be the point-wise log-likelihood. The $M$-unambiguity condition ensures that each coordinate of every vector in $\Delta_{c_S}^{\sigma,\epsilon}$ should be at least $\epsilon^M$, and hence the function $H$ is bounded on $\Delta_{c_S}^{\sigma,\epsilon}$. By Theorem 1 of (Jennrich, 1969), this ensures that $\sum_s H(s, \mathbf{r})$ converges uniformly to $\mathbb{E}_S[H(S, \mathbf{r})]$. According to (Vaart, 1998) (Theorem 5.7), the uniform convergence further ensures that $\Psi_\sigma(\widehat{\mathbf{r}}^*_{m_\mathsf{P}}) \to \mathbf{p}$ in probability as $m_\mathsf{P} \to \infty$. Since $\Psi_\sigma$ is invertible, this implies that $\widehat{\mathbf{r}}^*_{m_\mathsf{P}} \to \mathbf{r}$ in probability. $\square$

**Counterexample where invertibility fails to hold.** Consider the following transition function for binary labels ($\mathcal{Y} = \{0, 1\}$) and $M = 4$:

$$\sigma(y_1, y_2, y_3, y_4) = \begin{cases} 1, & \sum_{i=1}^{4} y_i \in \{1, 2, 4\} \\ 0, & \text{otherwise} \end{cases} \tag{25}$$

The $M$-unambiguity condition (Wang et al., 2023b) holds since $\sigma(0,0,0,0) \neq \sigma(1,1,1,1)$. On the other hand, the probability that the partial label equal to 1 can be expressed as:

$$\mathbb{P}(s=1) = r_1^4 + 6r_1^2 r_0^2 + 4r_1 r_0^3 = r_1^4 + 6r_1^2(1-r_1)^2 + 4r_1(1-r_1)^3 \qquad (26)$$

which is not an injection, see the plot of function $t \mapsto t^4 + 6t^2(1-t)^2 + 4t(1-t)^3$ in Figure 6.

## D    DETAILS FOR SECTION 4.2

### D.1    A NON-LINEAR PROGRAM FORMULATION

A straightforward idea that accommodates the requirements set in Section 4.2 is to reformulate (9) by (i) extending $\mathbf{P}$ (resp. $\mathbf{Q}$) to a tensor of size $n \times c \times M$ to store the scores (resp. pseudo-labels) of $M$-ary tuples of instances and (ii) modifying $U'$ so that the combinations of entries in $\mathbf{Q}$ corresponding to invalid label assignments are forced to have product equal to zero. However, modifying $U'$ in this way, we cannot employ Sinkhorn-like techniques as the one in (Lin et al., 2022), leaving us only with the option to employ non-linear[1] programming techniques to find $\mathbf{Q}$.

### D.2    DERIVING THE LINEAR PROGRAM IN (6)

Let $(x_{\ell,1}, \ldots, x_{\ell,M}, s_\ell)$ denote the $\ell$-th partial training sample, where $\ell \in [n]$. To derive the linear program in (6), we associate each partial label $s_\ell$ with a DNF formula $\Phi_\ell$, a process that is standard in the neurosymbolic literature (Xu et al., 2018; Tsamoura et al., 2021; Huang et al., 2021; Wang et al., 2023b). To ease the presentation, we describe how to compute $\Phi_\ell$. Let $\{\mathbf{y}_{\ell,1}, \ldots, \mathbf{y}_{\ell,R_\ell}\}$ be the set of vectors of labels in $\sigma^{-1}(s_\ell)$. We associate each prediction with a Boolean variable. Namely, let $q_{\ell,i,j}$ be a Boolean variable that becomes true when $x_{\ell,i}$ is assigned with label $j \in \mathcal{Y}$. Via associating predictions with Boolean variables, each $\mathbf{y}_{\ell,t}$ can be associated with a conjunction $\varphi_{\ell,t}$ over Boolean variables from $\{q_{\ell,i,j} | i \in [M], j \in [c]\}$. In particular, $q_{\ell,i,j}$ occurs in $\phi_{\ell,t}$ only if the $i$-th label in $\mathbf{y}_{\ell,t}$ is $j \in \mathcal{Y}$. Consequently, the training sample $(x_{\ell,1}, \ldots, x_{\ell,M}, s_\ell)$ is associated with the DNF formula $\Phi_\ell = \bigvee_{r=1}^{R_\ell} \varphi_{\ell,t}$ that encodes all vectors of labels in $\sigma^{-1}(s_\ell)$. We assume a canonical ordering over the variables occurring in $\varphi_{\ell,t}$, using $\varphi_{\ell,t,j}$ to refer to the $j$-th variable, and use $|\varphi_{\ell,t}|$ to denote the number of (unique) Boolean variables occurring $\varphi_{\ell,t}$. Based on the above, we have $\varphi_{\ell,t} = \bigwedge_{k=1}^{|\varphi_{\ell,t}|} \varphi_{\ell,t,k}$.

Similarly to (Srikumar & Roth, 2023), we use the Iverson bracket $[]$ to map Boolean variables to their corresponding integer ones, e.g., $[q_{\ell,i,j}]$, denotes the integer variable associated with the Boolean variable $q_{\ell,i,j}$.

We are now ready to construct linear program (6). Notice that the solutions of this program capture the label assignments that abide by $\sigma$, i.e., the labels assigned to each $(x_{\ell,1}, \ldots, x_{\ell,M})$ should be either of $\mathbf{y}_{\ell,1}, \ldots, \mathbf{y}_{\ell,R_\ell}$. The steps of the construction are (see (Srikumar & Roth, 2023)):

- (STEP 1) We translate each $\Phi_\ell$ into a CNF formula $\Phi'_\ell$ via the Tseytin transformation (Tseitin, 1983) to avoid the exponential blow up of the (brute force) DNF to CNF conversion.

- (STEP 2) We add the corresponding linear constraints out of each subformula in $\Phi'_\ell$.

Given $\Phi_\ell = \bigvee_{r=1}^{R_\ell} \varphi_{\ell,t}$, the Tseytin transformation associates a fresh Boolean variable $\alpha_{\ell,t}$ with each disjunction $\varphi_{\ell,t}$ in $\Phi_\ell$ and rewrites $\Phi_\ell$ into the following logically equivalent formula:

$$\Phi'_\ell := \underbrace{\bigvee_{t=1}^{R_\ell} \alpha_{\ell,t}}_{\Psi_\ell} \wedge \bigwedge_{t=1}^{R_\ell} (\alpha_{\ell,t} \leftrightarrow \varphi_{\ell,t}) \qquad (27)$$

After obtaining $\Phi'_\ell$, the construction of (6) proceeds as follows. The first inequality that will be added to (6) comes from formula $\Psi_\ell$. In particular, it will be the inequality $\sum_{t=1}^{R_\ell} [\alpha_{\ell,t}] \geq 1$, due

---

[1]Non-linearity comes from the KL term and by enforcing invalid label combinations to have product equal to zero.

to Constraint (3) from (Srikumar & Roth, 2023). The next inequalities come from the subformula $\bigwedge_{t=1}^{R_\ell}(\alpha_{\ell,t} \leftrightarrow \varphi_{\ell,t})$ from (27). The latter can be rewritten to the following two formulas:

$$\alpha_{\ell,t} \to \bigwedge_{k=1}^{|\varphi_{\ell,t}|} \varphi_{\ell,t,k} \tag{28}$$

$$\bigwedge_{k=1}^{|\varphi_{\ell,t}|} \varphi_{\ell,t,k} \to \alpha_{\ell,t} \tag{29}$$

According to Constraint (10) from (Srikumar & Roth, 2023), (28) and (29) are associated with the following inequalities:

$$-|\varphi_{\ell,t}|[\alpha_{\ell,t}] + \sum_{k=1}^{|\varphi_{\ell,t}|} [\varphi_{\ell,t,k}] \geq 0 \tag{30}$$

$$-\sum_{k=1}^{|\varphi_{\ell,t}|} [\varphi_{\ell,t,k}] + [\alpha_{\ell,t}] \geq (1 - |\varphi_{\ell,t}|) \tag{31}$$

which will also be added to the linear program.

Lastly, according to Constraint (5) from (Srikumar & Roth, 2023), we have an equality $\sum_{j=1}^{c}[q_{\ell,i,j}] = 1$, for each $\ell \in [n]$ and $i \in [M]$. The above equality essentially requires the scores of all pseudo-labels for a given instance $x_{\ell,i}$ to sum up to one. Finally, we require each pseudo-label $[q_{\ell,i,j}]$ to be in $[0,1]$, for each $\ell \in [n]$, $i \in [M]$, and $j \in [c]$.

Putting everything together, we have the following linear program:

**minimize** $\displaystyle\min_{(\mathbf{Q}_1,\ldots,\mathbf{Q}_m)} \sum_{i=1}^{M} \langle \mathbf{Q}_i, -\log(\mathbf{P}_i)\rangle,$

**subject to**
$$\begin{array}{llll}
\sum_{r=1}^{R_\ell}[\alpha_{\ell,t}] & \geq 1, & \ell \in [n], & \\
-|\varphi_{\ell,t}|[\alpha_{\ell,t}] + \sum_{k=1}^{|\varphi_{\ell,t}|}[\varphi_{\ell,t,k}] & \geq 0, & \ell \in [n], t \in [R_\ell] & \\
-\sum_{k=1}^{|\varphi_{\ell,t}|}[\varphi_{\ell,t,k}] + [\alpha_{\ell,t}] & \geq -1(1 - |\varphi_{\ell,t}|), & \ell \in [n], t \in [R_\ell] & \\
\sum_{j=1}^{c}[q_{\ell,i,j}] & = 1, & \ell \in [n], i \in [M] & \\
[q_{\ell,i,j}] & \in [0,1], & \ell \in [n], i \in [M], j \in [c] &
\end{array} \tag{32}$$

Program (6) results after adding to the above program constraints enforcing the hidden label ratios $\hat{\mathbf{r}}$.

**Example D.1.** *We demonstrate an example of* (6) *in the context of Example 1.1. We assume* $n = 2$. *We also assume that the partial labels* $s_1$ *and* $s_2$ *of the two partial samples in the batch are equal to* $0$ *and* $1$, *respectively. Due to the properties of the* max, *we have:*

$$\sigma^{-1}(0) = \{(0,0)\} \tag{33}$$

$$\sigma^{-1}(1) = \{(0,1),(1,0),(1,1)\} \tag{34}$$

*and formulas* $\Phi_1$ *and* $\Phi_2$ *are defined as:*

$$\Phi_1 = \underbrace{q_{1,1,0} \wedge q_{1,2,0}}_{\varphi_{1,1}} \tag{35}$$

$$\Phi_2 = \underbrace{q_{2,1,0} \wedge q_{2,2,1}}_{\varphi_{2,1}} \vee \underbrace{q_{2,1,1} \wedge q_{2,2,0}}_{\varphi_{2,2}} \vee \underbrace{q_{2,1,1} \wedge q_{2,2,1}}_{\varphi_{2,3}} \tag{36}$$

*The Tseytin transformation associates the fresh Boolean variables* $\alpha_{1,1}, \alpha_{2,1}, \alpha_{2,2},$ *and* $\alpha_{2,3}$ *to* $\varphi_{1,1},$ $\varphi_{2,1}, \varphi_{2,2},$ *and* $\varphi_{2,3}$, *respectively, and rewrites* $\Phi_1$ *and* $\Phi_2$ *to the following logically equivalent formulas:*

$$\Phi_1' = \alpha_{1,1} \wedge (\alpha_{1,1} \leftrightarrow \varphi_{1,1}) \tag{37}$$

$$\Phi_2' = (\alpha_{2,1} \vee \alpha_{2,2} \vee \alpha_{2,3}) \wedge (\alpha_{2,1} \leftrightarrow \varphi_{2,1}) \wedge (\alpha_{2,2} \leftrightarrow \varphi_{2,2}) \wedge (\alpha_{2,3} \leftrightarrow \varphi_{2,3}) \tag{38}$$

*The linear constraints that are added due to $\Phi_1'$ are:*

$$
\begin{aligned}
[\alpha_{1,1}] &\geq 1 \\
-|\varphi_{1,1}|[\alpha_{1,1}] + [q_{1,1,0}] + [q_{1,2,0}] &\geq 0 \\
-([q_{1,1,0}] + [q_{1,2,0}]) + [\alpha_{1,1}] &\geq -1(1 - |\varphi_{1,1}|)
\end{aligned}
\tag{39}
$$

*The linear constraints that are added due to $\Phi_2'$ are:*

$$
\begin{aligned}
[\alpha_{2,1}] + [\alpha_{2,2}] + [\alpha_{2,3}] &\geq 1 \\
-|\varphi_{2,1}|[\alpha_{2,1}] + [q_{2,1,0}] + [q_{2,2,1}] &\geq 0 \\
-|\varphi_{2,2}|[\alpha_{2,2}] + [q_{2,1,1}] + [q_{2,2,0}] &\geq 0 \\
-|\varphi_{2,3}|[\alpha_{2,3}] + [q_{2,1,1}] + [q_{2,2,1}] &\geq 0 \\
-([q_{2,1,0}] + [q_{2,2,1}]) + [\alpha_{2,1}] &\geq -1(1 - |\varphi_{2,1}|) \\
-([q_{2,1,1}] + [q_{2,2,0}]) + [\alpha_{2,2}] &\geq -1(1 - |\varphi_{2,2}|) \\
-([q_{2,1,1}] + [q_{2,2,1}]) + [\alpha_{2,3}] &\geq -1(1 - |\varphi_{2,3}|)
\end{aligned}
\tag{40}
$$

*Finally, the requirement that the pseudo-labels for each instance $x_{\ell,i}$ to sum up to one, for $\ell \in [2]$ and $i \in [2]$, and to lie in $[0,1]$ introduces the following linear constraints:*

$$
\begin{aligned}
\sum_{j=0}^{9}[q_{1,1,j}] &= 1 \\
\sum_{j=0}^{9}[q_{1,2,j}] &= 1 \\
\sum_{j=0}^{9}[q_{2,1,j}] &= 1 \\
\sum_{j=0}^{9}[q_{2,2,j}] &= 1 \\
[q_{1,i,j}] &\in [0,1], \quad i \in [2], j \in \{0, \ldots, 9\} \\
[q_{2,i,j}] &\in [0,1], \quad i \in [2], j \in \{0, \ldots, 9\}
\end{aligned}
\tag{41}
$$

## E  EXTENDED RELATED WORK

**Long-tail learning**. The term *long-tail learning* has been used to describe settings in which instances of some classes occur very frequently in the training set, with other classes being underrepresented. The problem has received considerable attention in the context of supervised learning with the proposed techniques operating either at training- or at testing-time. Techniques in the former category typically work by either reweighting the losses computed out of the original training samples (Cao et al., 2019; Tan et al., 2020; 2021) or by over- or under-sampling during training (Chawla et al., 2002; Buda et al., 2018). Techniques in the latter category work by modifying the classifier's output scores at testing-time and using the modified scores for classification (Kang et al., 2020; Peng et al., 2022), with LA being one of the most well-known techniques (Menon et al., 2021). LA modifies the classifier's scores at testing-time by subtracting the (unknown) gold ratios. In particular, the prediction of classifier $f$ given input $x$ is given by $arg\max_{j \in [c]} f^j(x) - \ln(r_j)$. Our empirical analysis shows that CAROT is more effective than LA.

Closest to our work is the study in (Peng et al., 2022). Unlike CAROT, the authors in (Peng et al., 2022) focus on single-instance PLL, assume that the marginal $\mathbf{r}$ is known, and use an optimal transport formulation (Peyré & Cuturi, 2020) to adjust the classifier's scores. In contrast, CAROT relies on the assumption that $\hat{\mathbf{r}}$ may be noisy, resorting to a robust optimal transport formulation (Le et al., 2021) to improve the classification accuracy in those cases.

**Partial Label Learning.** As discussed in (Wang et al., 2023b), MI-PLL is an extension to standard (single-instance) PLL (Cour et al., 2011; Lv et al., 2020; Feng et al., 2020). The observation that certain classes are harder to learn than others dates back to the work of (Cour et al., 2011) in the context of PLL. We are the first to provide such results for MI-PLL, unveiling also the relationship between $\sigma$ and class-specific risks.

**Long-tail PLL.** A few recently proposed papers lie in the intersection of long-tail learning and standard PLL, namely (Liu et al., 2021), RECORDS (Hong et al., 2023) and SOLAR (Wang et al., 2022), with the first one focusing on non-deep learning settings. RECORDS modifies the classifier's scores following the same basic idea with LA and uses the modified scores for training. However, it employs a momentum-updated prototype feature to estimate $\hat{\mathbf{r}}$. RECORDS's design allows it to be used with any loss function and to be trivially extended to support MI-PLL. Our empirical analysis shows that RECORDS is less effective than CAROT, leading to lower classification accuracy when the same loss is adopted during training.

SOLAR shares some similarities with LP. In particular, given single-instance PLL samples of the form $\{(x_1, S_1), \ldots, (x_n, S_n)\}$, where each $S_\ell \subseteq \mathcal{Y}$ is the partial label of the $\ell$-th PLL sample[2], SOLAR finds pseudo-labels $\mathbf{Q}$ by solving the following linear program:

$$\min_{\mathbf{Q} \in \Delta} \langle \mathbf{Q}, -\log(\mathbf{P}) \rangle \tag{42}$$

$$\text{s.t.} \quad \Delta := \left\{ [q_{\ell,j}]_{n \times c} \mid \mathbf{Q}^\mathsf{T} \mathbf{1}_n = \widehat{\mathbf{r}},\ \mathbf{Q}\mathbf{1}_c = \mathbf{c},\ q_{\ell,j} = 0 \text{ if } j \notin S_\ell \right\} \subseteq [0,1]^{n \times c}$$

Program (42) shows that the information of each partial label $S_\ell$ is strictly encoded into $\Delta$. To directly extend (42) to MI-PLL, we have two options:

- Use an $n \times c^M$ tensor $\mathbf{P}$ to store the model's scores, where cell $P[\ell, j_1, \ldots, j_c]$ stores the classifier's scores for the label vector $(j_1, \ldots, j_c)$ associated with the $\ell$-th training MI-PLL sample, for $1 \leq \ell \leq n$. However, that formulation would require an excessively large tensor, especially when $M$ gets larger.
- Use separate tensors $\mathbf{P}_1, \ldots, \mathbf{P}_M$ to represent the model's scores of the $M$ instances, and set for each $1 \leq \ell \leq n$, the product $P_1[\ell, j_1] \times \cdots \times P_M[\ell, j_c]$ to be 0 if $(j_1, \ldots, j_c)$ does not belong to $\sigma^{-1}(s_\ell)$. However, that formulation would lead to a non-linear program.

Neither choice is scalable for MI-PLL when $M$ is large[3]. To circumvent this issue, our work translates the information of the partial labels into linear constraints, leading to an LP formulation. Another difference between SOLAR and our work is that we provide Algorithm 1 to obtain ratio estimates, while SOLAR employs a window averaging technique to estimate $\mathbf{r}$ based on the model's own scores (Wang et al., 2022).

Finally, although CAROT also uses a linear programming formulation with a Sinkhorn-style procedure, it differs from SOLAR in that it adjusts the classifier's scores at testing-time rather than assigning pseudo-labels at training time.

**Constrained learning.** MI-PLL is closely related to constrained learning, in the sense that the predicted label vector $\mathbf{y}$ is subject to constraint $\sigma(\mathbf{y}) = s$. Training classifiers under constraints has been well studied in NLP (Steinhardt & Liang, 2015; Raghunathan et al., 2016; Peng et al., 2018; Mihaylova et al., 2020; Upadhyay et al., 2016; Wang et al., 2019a; Gupta et al., 2021). The work in (Roth & Yih, 2007) proposes a formulation for training under linear constraints; (Samdani et al., 2012) proposes a Unified Expectation Maximization (UEM) framework that unifies several constrained learning techniques including CoDL (Chang et al., 2007) and Posterior Regularization (Ganchev et al., 2010). In particular, (Mayhew et al., 2019) employs a conceptually similar idea by encoding prior information of the label frequency with a CoDL formulation to enhance partial label learning for the Named Entity Recognition (NER) task. The UEM framework was also adopted by (Li et al., 2023a) for neurosymbolic learning. Our LP formulation is orthogonal to the UEM. These two could be integrated though.

The theoretical framework for constrained learning in (Wang et al., 2023a) provides a generalization theory. The framework suggests that encoding the constraint during both the training and testing stages results in a better model compared to encoding it only during testing. This theory could be potentially extended to explain the advantage of LP-based methods and to characterize the necessary conditions for CAROT to improve model performance.

**Neurosymbolic learning and MI-PLL.** MI-PLL quite often arises in neurosymbolic learning (Manhaeve et al., 2018; Wang et al., 2019b; Dai et al., 2019; Yang et al., 2020; Tsamoura et al., 2021; Manhaeve et al., 2021b; Huang et al., 2021; Li et al., 2023a). However, none of the above works deals with learning imbalances.

There has been recent theoretical research on MI-PLL and related problems (Marconato et al., 2023; 2024; Wang et al., 2023b). The work in (Marconato et al., 2023; 2024) deals with the problem of characterizing and mitigating *reasoning shortcuts* in MI-PLL, under the prism of neurosymbolic learning. Intuitively, a reasoning shortcut is a classifier that has small partial risk, but high classification risk. For example, a reasoning shortcut is a classifier that may have a good accuracy on the overall task of returning the maximum of two MNIST digits, but low accuracy of classifying

---

[2] In standard PLL, each partial label is a subset of classes from $\mathcal{Y}$.

[3] Yet another non-linear formulation is presented in Section D based on RSOT (see Section A).

---

**Listing 1** Theory for the Smallest Parent benchmark.

```
land_transportation :- automobile, truck
other_transportation :- airplane, ship
transportation :- land_transportation, other_transportation
home_land_animal :- cat, dog
wild_land_animal :- deer, horse
land_animal :- home_land_animal, wild_land_animal
other_animal :- bird, frog
animal :- land_animal, other_animal
entity :- transportation, animal
```

---

MNIST digits. The work in (Marconato et al., 2023) showed that current neurosymbolic learning techniques are vulnerable to reasoning shortcuts. However, it offers no (class-specific) error bounds or any theoretical characterization of learning imbalances. The authors in (Wang et al., 2023b) were the first to propose necessary and sufficient conditions that ensure learnability of MI-PLL and to provide error bounds for a state-of-the-art neurosymbolic loss under approximations (Huang et al., 2021). Our theoretical analysis extends the one in (Wang et al., 2023b) by providing (i) class-specific risk bounds (in contrast to (Wang et al., 2023b), which only bounds $R(f)$) and (ii) stricter bounds for $R(f)$. In particular, as we show in Proposition 3.5, we can recover the bound from Lemma 1 in (Wang et al., 2023b) by relaxing (3).

**Other weakly-supervised setting.** Another well-known weakly-supervised learning setting is that of Multi-Instance Learning (MIL). In MIL, instances are not individually labelled, but grouped into sets which either contain at least one positive instance, or only negative instances and the aim is to learn a **bag classifier** (Sabato & Tishby, 2012; Sabato et al., 2010). In contrast, in MI-PLL, instances are grouped into tuples, with each tuple of instances being associated with a set of mutually exclusive label vectors, and the aim is to learn an *instance classifier*.

## F   Further experiments and details

**Why using SL and Scallop.** SL (Xu et al., 2018; Manhaeve et al., 2021a) has become the state-of-the-art approach to train deep classifiers in neurosymbolic learning settings. Training under SL requires computing a Boolean formula $\phi$ encoding all the possible label vectors in $\sigma^{-1}(s)$ for each partial training sample $(\mathbf{x}, s)$ and then computing the weighted model counting (Chavira & Darwiche, 2008) of $\phi$ given the softmax scores of $f$. SL has been effective in several tasks, including visual question answering (Huang et al., 2021), video-text retrieval (Li et al., 2023b), and fine-tuning language models (Li et al., 2024) and has nice theoretical properties (Wang et al., 2023b; Marconato et al., 2023). Due to its effectiveness, SL is now adopted by several neurosymbolic engines, DeepProbLog (Manhaeve et al., 2021a), namely, DeepProbLog's successors Manhaeve et al. (2021b), and Scallop (Huang et al., 2021; Li et al., 2023b).

In our empirical analysis we only use Scallop because it is the only engine at the moment offering a scalable SL implementation that can support our scenarios when $M \geq 3$. The requirement to compute $\sigma^{-1}(s)$ during training. Computing $\sigma^{-1}(s)$ is generally required by neurosymbolic learning techniques (Li et al., 2023a; Manhaeve et al., 2021a; Dai et al., 2019; Yang et al., 2020). This computation can become a bottleneck when the space of candidate label vectors grows exponentially, as it is the case in our MAX-$M$, SUM-$M$, and HWF-$W$ scenarios. As also experimentally shown by (Tsamoura et al., 2021; Wang et al., 2023b), the neurosymbolic techniques from (Manhaeve et al., 2021a;b; Dai et al., 2019; Li et al., 2023a; Yang et al., 2020) either time out after several hours while trying to compute $\sigma^{-1}(s)$, or lead to deep classifiers of much worse accuracy than Scallop. So, Scallop was the only engine that could support our experiments, balancing runtime with accuracy.

A further discussion on scalability issues in neurosymbolic learning can be found in Section 3.2 and 6 from (Wang et al., 2023b).

**Additional scenarios.** In addition, we carried experiments with two other scenarios that have been widely used as neurosymbolic benchmarks, SUM-$M$ (Manhaeve et al., 2018; Huang et al., 2021) and HWF-$M$ (Li et al., 2023a;b). SUM-$M$ is similar to MAX-$M$, however, instead of taking

the maximum, we take the sum of the gold labels. The HWF-$M$ scenario[4] was introduced in Li et al. (2020) and each training sample $((x_1, \ldots, x_M), s)$ consists of a sequence $(x_1, \ldots, x_M)$ of digits in $\{0, \ldots, 9\}$ and mathematical operators in $\{+, -, *\}$, corresponding to a valid mathematical expression, and $s$ is the result of the mathematical expression. As in SUM-$M$, the aim is to train a classifier for recognizing digits and mathematical operators. Notice that this benchmark is not i.i.d. since only specific types of input sequences are valid. The benchmark comes with a list of training samples, however, we created our own ones in order to introduce imbalances in the digits and operators distributions.

**Computational infrastructure.** The experiments ran on an 64-bit Ubuntu 22.04.3 LTS machine with Intel(R) Xeon(R) Gold 6130 CPU @ 2.10GHz, 3.16TB hard disk and an NVIDIA GeForce RTX 2080 Ti GPU with 11264 MiB RAM. We used CUDA version 12.2.

**Software packages.** Our source code was implemented in Python 3.9. We used the following python libraries: `scallopy`[5], `highspy`[6], `or-tools`[7], `PySDD`[8], `PyTorch` and `PyTorch vision`. Finally, we used part of the code[9] available from (Hong et al., 2023) to implement RECORDS and part of the code[10] available from (Wang et al., 2022) to implement the sliding window approximation for marginal estimation.

**Classifiers.** For MAX-$M$ and SUM-$M$ we used the MNIST CNN also used in (Huang et al., 2021; Manhaeve et al., 2018). For HWF-$M$, we used the CNN also used in (Li et al., 2023a;b). For Smallest Parent, we used the ResNet model also used in (Wang et al., 2022; Hong et al., 2023).

**Data generation.** To create datasets for MAX-$M$, Smallest Parent, SUM-$M$, and HWF-$M$ we adopted the approach followed in prior work, e.g., (Dai et al., 2019; Tsamoura et al., 2021; Wang et al., 2023b). In particular, to create each training sample, we draw instances $x_1, \ldots, x_M$ from MNIST or CIFAR-10 in an independent fashion. Then, we apply the transition $\sigma$ over the gold labels $y_1, \ldots, y_M$ to obtain the partial label $s$. To create datasets for HWF-$M$, we followed similar steps to the above, however, to make sure that the input vectors of images represent a valid mathematical expression, we split the training instances into operators and digits, drawing instances of digits for odd $i$s and instances of operators for even $i$s, for $i \in [M]$. Before dataset creation, we the images in HWF were split into training and testing ones with ratio 70%/30%, as the benchmark was not offering those splits. As we state in Section 5, to simulate long-tail phenomena (denoted as **LT**), we vary the imbalance ratio $\rho$ of the distributions of the input instances as in (Cao et al., 2019; Wang et al., 2022): $\rho = 0$ means that the hidden label distribution is unmodified and balanced. In each scenario, the test data follows the same distribution as the hidden labels in the training MI-PLL data, e.g., when $\rho = 0$, the test data is balanced; otherwise, it is imbalanced under the same $\rho$.

**Further details.** For the Smallest Parent scenarios, we computed SL and (6) using the whole pre-image of each partial label. For the MAX-$M$ scenarios, as the space of pre-images is very large, we only consider the top-1 proof (Wang et al., 2023b) both when running Scallop and in (6). For the Smallest Parent benchmark, we created the hierarchical relations shown in Listing 1 based on the classes of CIFAR-10.

To assess the robustness of our techniques, we focus on scenarios with high imbalances, large number of input instances, and few partial training samples. Table 3 shows results for SUM-$M$, for $M \in \{5, 6, 7\}$, $\rho = \{50, 70\}$, and $m_\mathsf{P} = 2000$. Table 4 shows results for HWF-$M$, for $M \in \{5, 6, 7\}$, $\rho = \{15, 50\}$, and $m_\mathsf{P} = 250$, while Table 5 shows results for the same experiment, but $m_\mathsf{P} = 1000$. In Tables 4 and 5, LP(ALG1) refers to running LP using the gold ratios– Algorithm 1 cannot be applied, as the data is not i.i.d. in this scenario. Tables 4 and 5 focuses on training-time mitigation. RECORDS was not considered as it led to substantially lower accuracy in the MAX-$M$ and Smallest Parent scenarios. Figure 7 shows the marginal estimates computed by Algorithm 1 for different scenarios. Last, Table 6 presents the full results for the MAX-$M$ scenarios. The tables follow the same notation with the ones in the main body of the paper.

---

[4]The benchmark is available at https://liqing.io/NGS/.

[5]https://github.com/scallop-lang/scallop (MIT license).

[6]https://pypi.org/project/highspy/ (MIT license).

[7]https://developers.google.com/optimization/ (Apache-2.0 license).

[8]https://pypi.org/project/PySDD/ (Apache-2.0 license).

[9]https://github.com/MediaBrain-SJTU/RECORDS-LTPLL (MIT license).

[10]https://github.com/hbzju/SoLar.

**Conclusions.** The conclusions that we can draw from Table 3, 4, 5 and Figure 7 are very similar to the ones that were drawn in the main body of our paper. When LP is adopted jointly with the estimates obtained via Algorithm 1, we can see that the accuracy improvements are substantial on multiple occasions. For example, in SUM-6 with $\rho = 50$, the accuracy increases from 67% under SL to 80% under LP(ALG1); in HWF-7 with $\rho = 15$, the accuracy increases from 37% under SL to 41% under LP(ALG1). The accuracy under LP(EMP) is lower than the accuracy under LP(ALG1) in SUM-$M$. We argue that this is because of the low quality of the empirical estimates of $\mathbf{r}$, a phenomenon that gets magnified due to the adopted approximations– recall that we run for SL and LP using the top-1 proofs, in order to make the computation tractable. The lower accuracy of LP(ALG1) for SUM-7 and $\rho = 70$ is attributed to the fact that the marginal estimates computed by Algorithm 1 diverge from the gold ones, see Figure 7. In fact, computing marginals for this scenario is particularly challenging due to the very large pre-image of $\sigma$ when $M = 7$, the high imbalance ratio ($\rho = 70$), and the small number of partial samples ($m_\mathsf{P} = 2000$). Tables 4 and 5 also suggest that SOLAR's empirical ratio estimation technique may harm LP's accuracy, supporting a claim that we also made in the main body of the paper, that *computing marginals for training-time mitigation is an important direction for future research.*

Figure 7 shows the robustness of Algorithm 1 in computing marginals. Figure 8 shows the hidden label ratios and the corresponding class-specific classification accuracies under the MAX-$M$ and the Smallest Parent scenarios for $\rho = 50$.

Table 3: Experimental results for SUM-$M$ using $m_\mathsf{P} = 2000$. Results over six runs.

| Algorithms | **LT** $\rho = 50$ | | | **LT** $\rho = 70$ | | |
|---|---|---|---|---|---|---|
| | $M = 5$ | $M = 6$ | $M = 7$ | $M = 5$ | $M = 6$ | $M = 7$ |
| SL | $82.28 \pm 15.87$ | $67.60 \pm 13.43$ | $68.42 \pm 25.66$ | $75.43 \pm 22.49$ | $79.60 \pm 19.36$ | $69.05 \pm 13.31$ |
| + LA | $81.74 \pm 16.27$ | $67.04 \pm 13.27$ | $68.33 \pm 25.61$ | $75.38 \pm 22.58$ | $79.47 \pm 19.49$ | $68.95 \pm 12.91$ |
| + CAROT | $82.21 \pm 15.94$ | $68.82 \pm 12.61$ | $69.54 \pm 24.46$ | $76.12 \pm 21.80$ | $80.47 \pm 18.37$ | $66.08 \pm 17.70$ |
| LP(EMP) | $75.31 \pm 23.49$ | $62.86 \pm 6.97$ | $62.89 \pm 34.47$ | $78.18 \pm 20.74$ | $64.66 \pm 33.95$ | $63.64 \pm 35.32$ |
| + LA | $74.94 \pm 23.86$ | $62.36 \pm 6.71$ | $62.55 \pm 34.81$ | $78.11 \pm 20.81$ | $64.02 \pm 34.66$ | $63.08 \pm 35.87$ |
| + CAROT | $72.19 \pm 17.50$ | $64.13 \pm 8.37$ | $65.26 \pm 32.24$ | $77.25 \pm 21.48$ | $66.36 \pm 27.43$ | $67.95 \pm 30.85$ |
| LP(ALG1) | $89.86 \pm 8.54$ | $80.10 \pm 18.45$ | $77.94 \pm 20.72$ | $91.64 \pm 7.62$ | $91.52 \pm 7.24$ | $63.79 \pm 12.97$ |
| + LA | $89.72 \pm 8.68$ | $79.43 \pm 19.15$ | $77.61 \pm 21.05$ | $91.66 \pm 7.60$ | $91.52 \pm 7.24$ | $63.70 \pm 12.87$ |
| + CAROT | $89.14 \pm 9.16$ | $78.85 \pm 19.55$ | $67.74 \pm 29.69$ | $91.29 \pm 7.86$ | $91.97 \pm 6.80$ | $67.06 \pm 9.78$ |

Table 4: Experimental results for HWF-$M$ using $m_\mathsf{P} = 250$. Results over six runs.

| Algorithms | **LT** $\rho = 15$ | | | **LT** $\rho = 50$ | | |
|---|---|---|---|---|---|---|
| | $M = 3$ | $M = 5$ | $M = 7$ | $M = 3$ | $M = 5$ | $M = 7$ |
| SL | $38.03 \pm 44.91$ | $44.83 \pm 5.22$ | $37.02 \pm 10.89$ | $39.94 \pm 46.83$ | $50.40 \pm 17.31$ | $36.83 \pm 20.94$ |
| LP(EMP) | $41.66 \pm 23.00$ | $44.16 \pm 7.33$ | $38.66 \pm 6.90$ | $45.56 \pm 39.70$ | $50.29 \pm 25.65$ | $34.38 \pm 16.60$ |
| LP(GOLD) | $48.31 \pm 26.72$ | $44.72 \pm 6.73$ | $41.06 \pm 8.05$ | $50.73 \pm 34.19$ | $51.63 \pm 14.00$ | $35.55 \pm 15.17$ |

Table 5: Experimental results for HWF-$M$ using $m_\mathsf{P} = 1000$. Results over six runs.

| Algorithms | **LT** $\rho = 15$ | | | **LT** $\rho = 50$ | | |
|---|---|---|---|---|---|---|
| | $M = 3$ | $M = 5$ | $M = 7$ | $M = 3$ | $M = 5$ | $M = 7$ |
| SL | $94.01 \pm 0.49$ | $95.34 \pm 0.14$ | $48.23 \pm 6.91$ | $27.42 \pm 25.62$ | $80.81 \pm 15.36$ | $83.87 \pm 13.00$ |
| LP(EMP) | $84.27 \pm 10.01$ | $84.86 \pm 10.80$ | $50.90 \pm 12.17$ | $49.26 \pm 45.98$ | $66.44 \pm 19.62$ | $47.04 \pm 8.58$ |
| LP(GOLD) | $94.39 \pm 0.27$ | $95.72 \pm 0.34$ | $55.73 \pm 6.12$ | $41.09 \pm 52.57$ | $81.28 \pm 14.43$ | $88.85 \pm 27.89$ |

Table 6: Experimental results for MAX-$M$ using $m_P = 3000$.

| Algorithms | Original $\rho = 0$ | | | LT $\rho = 5$ | | | LT $\rho = 15$ | | | LT $\rho = 50$ | | |
|---|---|---|---|---|---|---|---|---|---|---|---|---|
| | $M = 3$ | $M = 4$ | $M = 5$ | $M = 3$ | $M = 4$ | $M = 5$ | $M = 3$ | $M = 4$ | $M = 5$ | $M = 3$ | $M = 4$ | $M = 5$ |
| SL | 84.15 ± 11.92 | 73.82 ± 2.36 | 59.88 ± 5.58 | 55.48 ± 23.23 | 66.24 ± 1.22 | 55.13 ± 4.20 | 71.25 ± 4.48 | 66.98 ± 3.2 | 55.06 ± 5.21 | 66.74 ± 5.42 | 67.71 ± 11.58 | 55.74 ± 2.58 |
| + LA | 84.17 ± 11.95 | 73.82 ± 2.36 | 59.88 ± 5.58 | 55.48 ± 23.23 | 65.63 ± 1.75 | 55.13 ± 4.20 | 70.80 ± 4.52 | 66.98 ± 3.20 | 54.53 ± 5.74 | 66.57 ± 5.09 | 61.10 ± 3.95 | 52.47 ± 8.06 |
| + CAROT | 84.57 ± 11.50 | 73.08 ± 3.10 | 60.26 ± 5.20 | 56.52 ± 21.70 | 66.70 ± 0.76 | 55.91 ± 3.42 | 74.95 ± 3.45 | 67.44 ± 2.74 | 55.80 ± 4.47 | 68.16 ± 4.00 | 68.25 ± 6.14 | 57.29 ± 14.17 |
| RECORDS | 85.56 ± 7.25 | 75.11 ± 0.77 | 59.43 ± 6.61 | 77.98 ± 3.13 | 65.85 ± 0.62 | 55.07 ± 4.24 | 55.47 ± 20.45 | 53.34 ± 16.66 | 52.40 ± 7.95 | 70.20 ± 7.65 | 66.05 ± 13.90 | 59.93 ± 4.86 |
| + LA | 87.63 ± 5.11 | 75.11 ± 0.77 | 59.28 ± 6.76 | 77.98 ± 3.13 | 65.43 ± 0.87 | 54.40 ± 4.44 | 54.90 ± 20.16 | 54.46 ± 15.54 | 51.25 ± 9.09 | 70.09 ± 7.26 | 65.78 ± 14.18 | 59.93 ± 4.86 |
| + CAROT | 90.97 ± 2.03 | 75.94 ± 0.91 | 60.45 ± 7.78 | 78.31 ± 4.00 | 67.57 ± 1.74 | 55.46 ± 3.94 | 54.32 ± 21.85 | 62.74 ± 8.14 | 55.85 ± 4.61 | 71.46 ± 6.4 | 71.25 ± 8.70 | 63.64 ± 5.92 |
| LP(EMP) | 94.97 ± 1.32 | 77.86 ± 4.22 | 55.27 ± 11.27 | 80.15 ± 1.69 | 70.73 ± 1.85 | 56.28 ± 2.03 | 75.83 ± 5.26 | 69.67 ± 5.47 | 59.25 ± 7.27 | 77.16 ± 3.46 | 70.06 ± 10.73 | 56.79 ± 1.58 |
| + LA | 94.69 ± 1.60 | 77.91 ± 4.16 | 55.34 ± 11.19 | 80.08 ± 1.55 | 70.54 ± 1.82 | 55.31 ± 3.27 | 75.77 ± 5.32 | 68.92 ± 3.96 | 58.49 ± 5.74 | 77.1 ± 3.52 | 69.76 ± 10.31 | 56.81 ± 1.56 |
| + CAROT | 95.07 ± 1.20 | 75.53 ± 7.42 | 53.07 ± 12.99 | 80.29 ± 2.33 | 70.88 ± 2.22 | 57.85 ± 4.05 | 76.38 ± 4.72 | 69.74 ± 5.51 | 59.56 ± 8.14 | 77.58 ± 3.04 | 70.11 ± 10.34 | 57.09 ± 1.90 |
| LP(ALG1) | 96.09 ± 0.41 | 78.34 ± 4.80 | 59.91 ± 6.63 | 78.56 ± 1.52 | 69.71 ± 0.03 | 57.61 ± 3.09 | 74.51 ± 9.13 | 69.14 ± 1.82 | 56.81 ± 3.74 | 72.23 ± 11.49 | 69.28 ± 11.78 | 63.67 ± 7.04 |
| + LA | 95.81 ± 0.74 | 78.97 ± 4.09 | 59.98 ± 6.56 | 78.48 ± 1.53 | 69.71 ± 0.03 | 57.47 ± 3.09 | 74.26 ± 9.06 | 68.73 ± 2.23 | 56.37 ± 3.13 | 72.23 ± 11.49 | 69.21 ± 11.86 | 63.67 ± 7.04 |
| + CAROT | 96.13 ± 0.38 | 80.78 ± 2.36 | 59.71 ± 6.35 | 78.93 ± 1.85 | 70.32 ± 0.86 | 57.62 ± 3.08 | 77.05 ± 7.00 | 69.19 ± 1.81 | 59.76 ± 7.24 | 74.82 ± 10.18 | 74.30 ± 7.54 | 64.39 ± 6.43 |

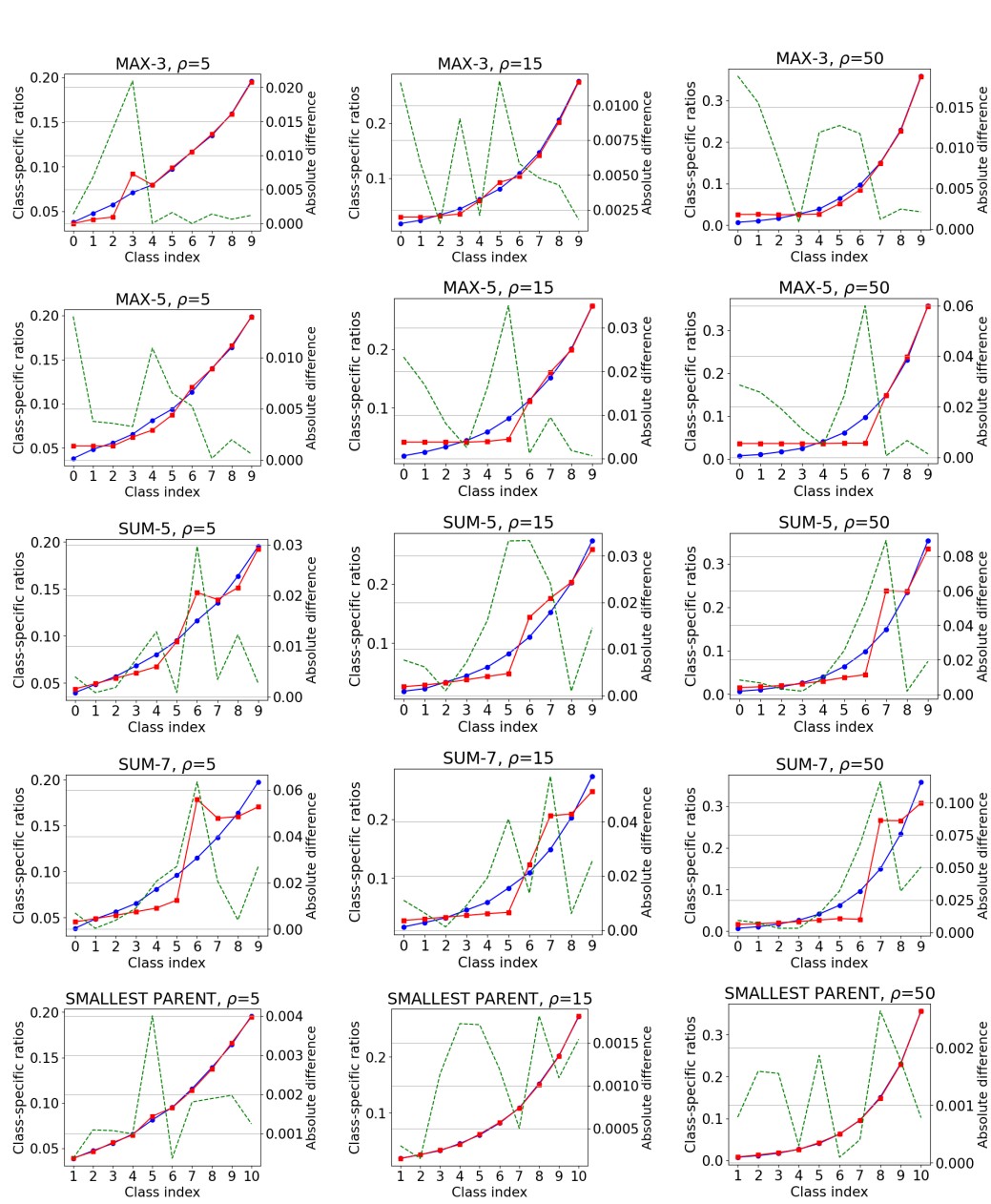

Figure 7: Accuracy of the marginal estimates computed by Algorithm 1 for different scenarios. Blue denotes the gold ratios, red the estimated ones, and green the absolute difference between the gold and estimated ratios.

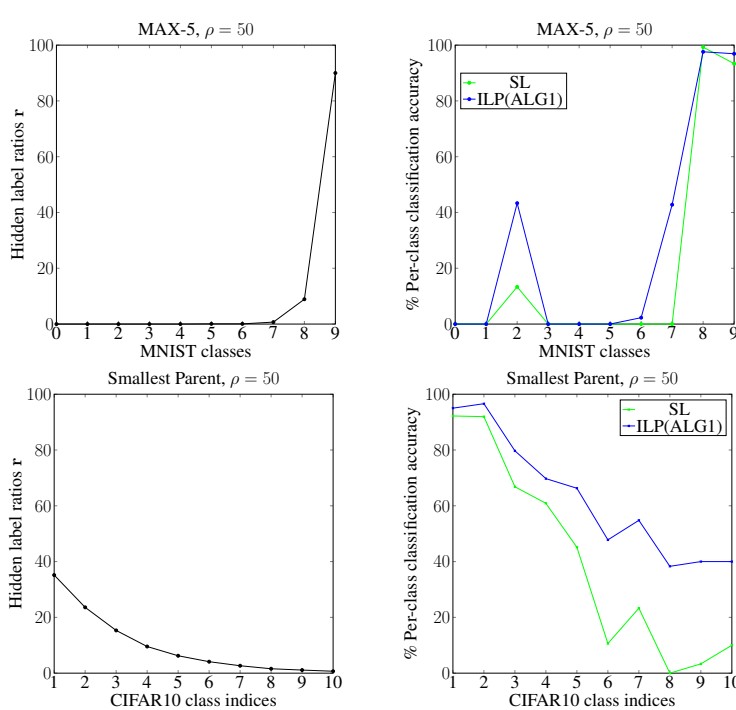

Figure 8: (Up left) hidden label ratios **r** for MAX-5 with $\rho = 50$. (Up right) Class-specific classification accuracies under SL and ILP(ALG1) for MAX-5 with $\rho = 50$. (Down left) hidden label ratios **r** for Smallest parent with $\rho = 50$. (Down right) Corresponding class-specific classification accuracies under SL and ILP(ALG1) for Smallest parent with $\rho = 50$.

Table 7: The notation in the preliminaries and the theoretical analysis.

| Supervised learning notation | |
|---|---|
| $1\{\cdot\}$ | Indicator function |
| $[n] := \{1, \ldots, n\}$ | Set notation |
| $\mathcal{X}, \mathcal{Y} = [c]$ | Input instance space and label space |
| $x, y$ | Elements from $\mathcal{X}$ and $\mathcal{Y}$ |
| $X, Y$ | Random variables over $\mathcal{X}$ and $\mathcal{Y}$ |
| $\mathcal{D}, \mathcal{D}_X, \mathcal{D}_Y$ | Joint distribution of $(X, Y)$ and marginals of $X$ and $Y$ |
| $r_j = \mathbb{P}(Y = j)$ | probability of occurrence (or ratio) of label $j \in \mathcal{Y}$ in $\mathcal{D}$ |
| $\mathcal{D}_Y := \mathbf{r} = (r_1, \ldots, r_c)$ | Marginal of $Y$ |
| $\Delta_c$ | Space of probability distributions over $\mathcal{Y}$ |
| $f : \mathcal{X} \to \Delta_c$ | Scoring function |
| $f^j(x)$ | Score of $f$ upon $x$ for class $j \in \mathcal{Y}$ |
| $[f] : \mathcal{X} \to \mathcal{Y}$ | Argmax classifier induced by $f$ |
| $\mathcal{F}, [\mathcal{F}]$ | Space of scoring functions and corresponding space of classifiers |
| $d_{[\mathcal{F}]}$ | Natarajan dimension of $[\mathcal{F}]$ |
| $L(y', y) := 1\{y' \neq y\}$ | Zero-one loss given $y, y' \in \mathcal{Y}$ |
| $R(f)$ | Zero-one risk of $f$ |
| $R_j(f) := P([f](x) \neq j \mid Y = j)$ | Risk of $f$ for the $j$-th class in $\mathcal{Y}$ |
| $D(\mathbf{A})$ | The diagonal matrix that shares the same diagonal with square matrix $\mathbf{A}$ |

| MI-PLL notation | |
|---|---|
| $M > 0$ | Number of input instances per MI-PLL sample |
| $\mathbf{x} = (x_1, \ldots, x_M), \mathbf{y} = (y_1, \ldots, y_M)$ | Vector of input instances and their (hidden) gold label |
| $\mathcal{S} = \{a_1, \ldots, a_{c_S}\}$ | Space of $c_S$ partial labels |
| $S$ | Random variable over $\mathcal{S}$ |
| $\sigma : \mathcal{Y}^M \to \mathcal{S}$ | Transition function (known to the learner) |
| $s = \sigma(\mathbf{y})$ | Partial label |
| $\sigma^{-1}(s)$ | Pre-image of $s$, i.e., set of all vectors $\mathbf{y} \in \mathcal{Y}^M$ s.t. $\sigma(\mathbf{y}) = s$ |
| $(\mathbf{x}, s)$ | Partial sample |
| $\mathcal{D}_{\mathsf{P}}$ | Distribution of partial samples over $\mathcal{X}^M \times \mathcal{S}$ |
| $\mathcal{D}_{\mathsf{P}_S}$ | Marginal of $S$ |
| $\mathcal{T}_{\mathsf{P}}$ | Set of $m_{\mathsf{P}}$ partial samples |
| $[f](\mathbf{x})$ | Short for $([f](x_1), \ldots, [f](x_M))$ |
| $L_\sigma(\mathbf{y}, s) := L(\sigma(\mathbf{y}), s)$ | Zero-one partial loss subject to $\sigma$ |
| $R_{\mathsf{P}}(f; \sigma) := E_{(X_1, \ldots, X_M, S) \sim \mathcal{D}_{\mathsf{P}}}[L_\sigma(([f](\mathbf{X})), S)]$ | Zero-one partial risk subject to $\sigma$ |
| $\widehat{R}_{\mathsf{P}}(f; \sigma, \mathcal{T}_{\mathsf{P}})$ | Empirical zero-one partial risk subject to $\sigma$ given set $\mathcal{T}_{\mathsf{P}}$ of partial samples |

| Notation in Section 3 | |
|---|---|
| $\mathbf{1}_n, \mathbf{0}_n$ | All-one and all-zero vectors |
| $\mathbf{I}_n$ | Identity matrix of size $n \times n$ |
| $\mathbf{e}_j$ | $c$-dimensional one-hot vector, where the $j$-th element is one |
| $\mathbf{H}(f)$ | $c \times c$ matrix where the $(i, j)$ cell is the probability of $f$ classifying an instance with label $i \in \mathcal{Y}$ to $j \in \mathcal{Y}$. |
| $\mathbf{h}(f) := \mathrm{vec}(\mathbf{H}(f))$ | Vectorization of $\mathbf{H}(f)$ |
| $\mathbf{w}_j := \mathrm{vec}(\mathbf{W}_j)$ | Vectorization of matrix $\mathbf{W}_j := (\mathbf{1}_c - \mathbf{e}_j)\mathbf{e}_j^\top$, where $j \in \mathcal{Y}$ |
| $\mathbf{\Sigma}_{\sigma, \mathbf{r}}$ | Symmetric matrix in $R^{c^2 \times c^2}$ depending on $\sigma$ and $\mathbf{r}$ |
| $\Phi_{\sigma, j}(R_{\mathsf{P}}(f; \sigma))$ | Optimal solution to program (3) and upper bound to $R_j(f)$ |
| $\tilde{R}_{\mathsf{P}}(f; \sigma, \mathcal{T}_{\mathsf{P}}, \delta)$ | Generalization bound of $R_{\mathsf{P}}(f; \sigma)$ for probability $1 - \delta$ |

Table 8: The notation used in our proposed algorithms.

| Notation in Section 4.1 | |
|---|---|
| $p_j := \mathbb{P}(S = a_j)$ | Probability of occurrence (or ratio) of $a_j \in \mathcal{S}$ in $\mathcal{D}_\mathsf{P}$ |
| $P_\sigma$ | System of polynomials $[p_j]_{j \in [c_S]}^\mathsf{T} = [\sum_{(y_1,\ldots,y_M) \in \sigma^{-1}(a_j)}]_{j \in [c_S]}^\mathsf{T}$ |
| $\Psi_\sigma$ | Mapping of each $r_j \in \mathcal{Y}$ to its solution in $P_\sigma$, assuming $\mathbf{p}$ is known |
| $\widehat{\mathbf{r}}, \widehat{\mathbf{p}}$ | Estimates of $\mathbf{r}$ and $\mathbf{p}$ |
| $\bar{p}_j := \sum_{k=1}^{m_\mathsf{P}} \mathbb{1}\{s_k = a_j\}/m_\mathsf{P}$ | Estimate of $p_j$ given partially labeled dataset $\mathcal{T}_\mathsf{P}$ |

| Notation in Section 4.2 | |
|---|---|
| $n > 0$ | Size of each batch of partial samples |
| $i$ | Index over $[M]$ |
| $j$ | Index over $[c]$ |
| $\ell$ | Index over $[n]$ |
| $(x_{\ell,1}, \ldots, x_{\ell,M}, s_\ell)$ | $\ell$-th partial training sample in the input batch |
| $R_\ell$ | Size of $\sigma^{-1}(s_\ell)$ |
| $t$ | Index over $[R_\ell]$ |
| $\mathbf{P}_i$ | Matrix in $[0,1]^{n \times c}$, where $P_i[\ell, j] = f^j(x_{\ell,i})$ |
| $\mathbf{Q}_i$ | Matrix in $[0,1]^{n \times c}$, where $Q_i[\ell, j]$ is the pseudo-label assigned with label $j \in \mathcal{Y}$ for instance $x_{\ell,i}$ |
| $q_{\ell,i,j}$ | A Boolean variable that is true if $x_{\ell,i}$ is assigned with label $j \in \mathcal{Y}$ and false otherwise |
| $\varphi_{\ell,t}$ | Conjunction over the $q_{\ell,i,j}$ Boolean variables that encodes the $t$-th label vector in $\sigma^{-1}(s_\ell)$ |
| $\Phi_\ell = \varphi_{\ell,1} \vee \cdots \vee \varphi_{\ell,R_\ell}$ | DNF formula encoding the label vectors in $\sigma^{-1}(s_\ell)$ |
| $\alpha_{\ell,t}$ | A fresh Boolean variable associated with each $\varphi_{\ell,t}$ by the Tseytin transformation |

| Notation in Section 4.3 | |
|---|---|
| $n > 0$ | Size of each batch of test input instances from $\mathcal{X}$ |
| $\mathbf{P}$ | Matrix in $R^{n \times c}$ of the $f$'s scores on the test instances of the input batch |
| $\mathbf{P}'$ | Matrix in $R^{n \times c}$ storing the CAROT's adjusted scores for $\mathbf{P}$ |
| $H(\mathbf{P}')$ | Entropy of $\mathbf{P}'$ |
| $\eta, \tau > 0$ | Parameters of robust semi-constrained optimal transport problem (Le et al., 2021) |

