# OpenReview forum: "On Characterizing and Mitigating Imbalances in Multi-Instance Partial Label Learning"
_ICLR.cc/2025/Conference — Submitted to ICLR 2025_

### Official Review · Reviewer_QVdr · 2024-10-29

**Soundness:** 3
**Presentation:** 2
**Contribution:** 2
**Rating:** 6
**Confidence:** 4

**Summary:**

This paper explores a weakly-supervised learning framework that integrates aspects of partial label learning and partial label learning. It presents theoretical contributions by establishing class-specific risk bounds under minimal assumptions and demonstrating the significant impact of transition functions on learning imbalances. Additionally, the paper introduces practical algorithms for estimating hidden label distributions and mitigating learning imbalances at both training and testing phases, achieving performance improvements of up to 14% over strong baselines.

**Strengths:**

1. This is an theoritically-solid paper - the proofs and theorems appears to be correct and sound.

2. This paper have proposes several notable and interesting theoritical properties, which could be benefical for the future development of MI-PLL.

3. This paper introduces two new algorithms to mitigate imbalances at training and inference time.

4. The proposed method obtained competitive performances comparing with its closest counterparts.

**Weaknesses:**

1. This paper becomes challenging to comprehend as the authors have not provided clear definitions of a partial label. Specifically, by examining Example 1.1 alone, the authors' characterization of partial labeling seems to differ from existing work [1].

2. It appears that the authors' proposed method is somewhat orthogonal to their theoretical results, suggesting a divergence between practice and theory.

3. While it is reasonable to assume that imbalances in Multi-Instance Partial Label Learning (MI-PLL) arise not only from label distribution but also from the partial labeling process, the theoretical results of this paper seem to only consider the imbalances caused by partial labeling. This limitation potentially restricts the flexibility and applicability of the theoretical findings.

[1] Learning from Partial Labels, JMLR 2011.

**Questions:**

1. Is Equation 1 introduced by the authors or derived from previous works? I find the mathematical connections a bit challenging to grasp, particularly as it doesn’t seem to directly correspond to the definition of partial risk outlined by Cour et al. If this is an original contribution from your team, shouldn’t Equation 1 be presented as a lemma with accompanying proofs to substantiate its validity?

2. I'm not completely certain, but it appears that Proposition 3.3 suggests the dependency of the empirical partial risk on the imbalance of partial labels, while your measure of complexity (e.g., the Natarajan dimension) remains unaffected by the level of imbalance. If this interpretation is accurate, does this imply that minimizing the partial risk is invariably advantageous in imbalanced MI-PLL scenarios, regardless of the extent of imbalance?

---

> ### Author Response · Authors · 2024-11-21
>
> > Weakness 1
>
> Thanks for the comment. Our definition of partial labels differs slightly from the one in (Cour et al., 2011) since we are considering a multi-instance extension of PLL. We still refer to it as a PLL problem because, in our case, each partial label can still be represented as a superset of candidate hidden labels, consistent with the definition in (Cour et al., 2011).
>
> As we state at the beginning of Section 2, we used this definition to align with the notation and definitions in (Wang et al., 2023b).
>
> To improve readability, we uploaded a revised version of our work in which we include tables summarizing our notation (please see Table 7 and 8 in the appendix).
>
> > Weakness 2
>
> Our analysis focuses on \emph{theoretically characterizing} learning imbalances in MI-PLL, by deriving class-specific error bounds using techniques common in learning theory. Instead, our proposed algorithms aim to mitigate the learning imbalances.
>
> This is an analogy to the (independent) work of (Cour et al., 2011) and (Wang et al., 2022; Hong et al., 2023): (Cour et al., 2011)  focuses on a theoretical characterization of class-specific risks for standard PLL, while (Wang et al., 2022; Hong et al., 2023) focuses on imbalanced learning in standard PLL.
>
> So, the contributions of the theoretical and algorithmic parts are indeed orthogonal, but this does not mean that the practice diverges from theory, kindly disagreeing with the reviewer. Instead, \textbf{the practice is motivated} by theoretical algorithms to solve a problem nobody else has studied before.
>
> > Weakness 3
>
> In fact, our theoretical part considers both the imbalances caused by partial labeling and the label distribution.
> The label distribution $\mathbf r$ could impact the generalization bounds by affecting the solution of the optimization program (3), which plays a key part in our theory.
> For example, Figure 2, illustrates the impact of the hidden label distribution (uniform vs non-uniform) on the imbalance, even when the partial labeling process is the same in the two subplots.
> In our paper, we emphasized the implication on the imbalances caused by partial labeling, since it is a novel phenomenon in MI-PLL.
>
> > Question 1
>
> Equation 1 is a novel but straightforward rewriting of the zero-one partial risk from Section 2.
> To derive it, we enumerate all the 4-tuples $(i,j,{i'},{j'}) \in \mathcal{Y}^4$ where $i,j$ are the gold hidden labels and ${i'},{j'}$ are the predicted labels so that this prediction leads to a mistake in the output partial labels.
> The partial risk is essentially the sum of the probabilities of those events.    The term ${1\lbrace\sigma(i, j) \ne \sigma({i'}, {j'})\rbrace}$ indicates that the partial label is mis-classified.
> The term $H_{ii'}(f)H_{jj'}(f)$ encodes the probability of the tuple $(i,j,{i'},{j'})$ by definition.
>
> We have also added this clarification below Equation (1) in the revised version.
>
> > Question 2
>
> The interpretation is correct since the Natarajan dimension only measures the complexity of the model, i.e., the set of all candidate classifiers.
> In this sense, this measure remains unaffected by the level of imbalance.
> However, it does not imply that minimizing the partial risk is more advantageous because it does not fully describe the difficulty of learning. The difficulty of learning is characterized by the generalization bounds (Proposition 3.3), which do depend on the imbalance itself.

---

> > ### Comment · Reviewer_QVdr · 2024-11-24
> >
> > I appreciate the detailed rebuttal from the authors; here are my responses:
> >
> > W1. Thank you for highlighting this difference. However, my concern persists: it appears that the MI-PLL setup in this paper is relatively novel and not widely adopted, as evidenced by the limited references. This potentially suggests that the scope or the applicability of this problem is narrow.
> >
> > W2. While I partially agree with the authors, I question the broader utility of the theoretical results derived in this paper. If these results are indeed meaningful, should they not offer practical insights that could inform the design of empirical heuristics? Moreover, the orthogonality you describe seems to undermine the motivation behind your proposed algorithms. Currently, I see little motivation or justification for your methods. I believe that existing techniques for addressing long-tailed learning could be applicable in this scenario as well.
> >
> > Q2: Regarding Proposition 3.3, my understanding is that only the partial risk is dependent on the level of imbalances. Is there any other terms in Proposition 3.3 that also depends on the level of imbalance?

---

> ### Author Response · Authors · 2024-11-24
>
> Thank you for your prompt reply. Our responses to your further concerns are provided in the following thread.
>
> > W1. The MI-PLL setup is relatively novel and not widely adopted, as evidenced by the limited references
>
> In fact, our MI-PLL formulation is receiving close attention lately in the context of neurosymbolic learning (Manhaeve et al., 2018; Wang et al., 2019b; Dai et al., 2019; Tsamoura et al., 2021; Huang et al., 2021; Li et al., 2023a; Marconato et al., 2023; 2024). There, as we state in lines 047--053 of our submission, MI-PLL has been very beneficial in a variety of applications, including visual question answering (Huang et al., 2021), video-text retrieval (Li et al., 2023b), and fine-tuning language models (Zhang et al., 2023; Li et al., 2024), e.g., by exploiting a few simple symbolic commonsense rules, the authors of (Huang et al., 2021) managed to build neurosymbolic architectures that had substantially higher accuracy than state-of-the-art deep networks for visual question answering, see Table 2 from (Huang et al., 2021).
>
> The above strong practical results, the potential of the broader neurosymbolic learning setting, and the lack of previous theoretical analysis suggest that MI-PLL is a setting that is worth further exploring.
>
> However, **it is not only neurosymbolic learning where MI-PLL arises**. As we state in lines 047--048 of our submission, MI-PLL has been a topic of active research in NLP (Steinhardt \& Liang, 2015; Raghunathan et al., 2016; Peng et al., 2018; Wang et al., 2019a; Gupta et al., 2021).
>
> Furthermore, and as already stated in lines 034--041 of our submission, **MI-PLL encompasses two other well-known learning settings**:
>
> - **partial label learning (PLL)** (Cour et al., 2011; Cabannes et al., 2020; Lv et al., 2020; Seo \& Huh, 2021; Wen et al., 2021; Xu et al., 2021; Yu et al., 2022; Wang et al., 2022; Hong et al., 2023), where each training instance is associated with a set of candidate labels
> - **latent structural learning** (Steinhardt \& Liang, 2015; Raghunathan et al., 2016; Zhang et al., 2020), i.e., learning classifiers subject to a transition function $\sigma$ that constraints their outputs. The authors of (Wang et al., 2023b) have shown how MI-PLL generalizes PLL in their appendix.
>
> Based on the above, we believe that MI-PLL has a broader applicability.
>
> > If these results are indeed meaningful, should they not offer practical insights that could inform the design of empirical heuristics?
>
> Our theory mainly serves as the motivation for our problem. Our results are error bounds suggesting a phenomenon nobody else has observed in the relevant literature: that learning in MI-PLL is inherently imbalanced. These error bounds motivated us to develop new techniques for imbalance-aware learning. In fact, error bounds derived from learning theory typically do not lead to heuristics as you suggest, just as standard learning theory (e.g., VC-dimension) does not directly tell what algorithms should be applied to a specific problem. Nevertheless, these theories offer an important measure of the difficulty/complexity of the ML problem itself.
>
> > Moreover, the orthogonality you describe seems to undermine the motivation behind your proposed algorithms.
>
> As we already stated at the beginning of Section 4, enforcing the class priors gives more importance to the minority classes at training time (as ILP does in Section 4.2) and encourages $f$ to predict minority classes at testing time (as CAROT does in Section 4.3). We also discussed these points in our earlier responses.
>
> Our empirical results in Section 5 and Appendix F show that our techniques lead to classifiers with up to 14\% higher accuracy. Figure 8 in the appendix shows concrete examples of the effectiveness of our training-time imbalance mitigation techniques.
> For example, for $M=5$ and $\rho=50$, see the upper left part of Figure 8.
> In this scenario, for the baseline model (row SL in Table 1, for $M=5$ and $\rho=50$), the class-specific classification accuracies are:
>
> 0\%, 0\%, 13.33\%, 0\%, 0\%, 0\%, 0\%, 0\%, 99.28\%, 93.35\%.
>
> In contrast, for our training-time mitigation technique (row ILP(ALG1) in Table 1, for $M=5$ and $\rho=50$), class-specific classification accuracies are:
>
> 0\%, 0\%, 43,33\%, 0\%, 0\%, 0\%, 2.29\%, 42.80\%, 97.61\%, 96.91\%,
>
> substantially improving the classification accuracy for the MNIST digits 2, 6, and 7.
> Similar conclusions are drawn from the Smallest parent scenario, please see the lower part of Figure 8 in the appendix.
>
> Hence, our error bounds provided a good motivation to study imbalance-aware learning in MI-PLL and develop practical algorithms that lead to classifiers with substantially higher accuracy.

---

> ### Author Response · Authors · 2024-11-24
>
> (contd.)
>
> > I believe that existing techniques for addressing long-tailed learning could be applicable in this scenario as well.
>
> As we already discussed in lines 076--083 and lines 287--294, **it is not straightforward to apply existing long tail learning techniques in MI-PLL.**
>
> Starting from the problem of mitigating imbalances at training time, **differently** for existing techniques in supervised and semi-supervised learning, in MI-PLL
>
> - The training samples are $M$-ary tuples of instances of the form ${(x_1,\dots,x_M)}$
> - The label predictions (including **pseudo-labels**, which play an important role in many imbalanced learning algorithms) must additionally abide by the constraints coming from $\sigma$ and the partial labels, e.g., when $s=1$ in Example~1.1, then the only valid label assignments for ${(x_1,x_2)}$ are (1,1), (0,1) and (1,0).
>
> Furthermore, we show that **we cannot straightforwardly extend existing PLL techniques for mitigating imbalances at training time in the context of MI-PLL**, as that would lead to non-convex optimization problems, see Appendix D and E for further discussion. Beyond the above, a comparison against RECORDS (Hong et al., 2023), the most recent state-of-the-art technique for imbalance mitigation at training time in standard PLL, shows that RECORDS may harm the accuracy of the baseline model when applied in the context of MI-PLL, see Section 5 and Appendix F, suggesting again that a **straightforwardly application of the state-of-the-art techniques may be problematic**.
>
> Moving to testing-time mitigation, again, **we cannot straightforwardly extend existing supervised, semi-supervised or PLL learning techniques for mitigating imbalances at testing time in the context of MI-PLL** for two reasons: (i) most existing testing-time mitigation algorithms (e.g., (Menon et al., 2021)) modify a model's scores at the level of individual instances and (ii) strictly enforce the class prior ${\mathbf{r}}$. Both (i) and (ii) are problematic in MI-PLL as already discussed in lines 347-353 of our submission.
>
> Lastly, estimating the marginal of the hidden labels $\mathbf{r}$ in MI-PLL is not as straightforward as in standard supervised learning. We are the first to provide a statically consistent technique for estimating the marginal $\mathbf{r}$ from MI-PLL training samples. As we show in Section 5, employing our marginals further boosts the effectiveness of our algorithms for mitigating imbalances at training and testing time.
>
> Based on the above, we conclude that existing long-tail learning techniques cannot be directly applied to our setting and new techniques need to be developed specifically for MI-PLL.
>
> > Q2: Regarding Proposition 3.3, my understanding is that only the partial risk is dependent on the level of imbalances. Is there any other terms in Proposition 3.3 that also depends on the level of imbalance?
>
> No, the imbalance only impacts the empirical partial risk, as it does not change the complexity of the classifier space itself since the complexity is independent of the training data distribution.

---

### Official Review · Reviewer_98Rq · 2024-10-30

**Soundness:** 2
**Presentation:** 1
**Contribution:** 2
**Rating:** 3
**Confidence:** 4

**Summary:**

This paper addresses the problem of learning imbalances in Multi-Instance Partial Label Learning (MI-PLL), where the true labels are hidden and supervision is provided via a transition function. The authors derive theoretical bounds for class-specific risks, and propose methods to estimate hidden label distributions and introduce algorithms to mitigate these imbalances during both training and testing.

**Strengths:**

1. The paper provides a theoretical contribution by analyzing learning imbalances in MI-PLL, offering a deeper understanding of the problem.

2. The authors introduce novel algorithms for both training and testing phases, with empirical evidence.

**Weaknesses:**

1. The notation used in this paper is quite convoluted and makes it really hard to follow. I spent a lot of time carefully reading Section 2, but I still don’t have a clear understanding of what exactly the MI-PLL setting is or how it’s supposed to be used in practice.I would recommend reconsidering whether certain notations, e.g., $r_y = r_{l_j} = r_j$ , are essential in the manuscript. Additionally, including a notation table may be helpful for readers to quickly reference symbols, which would enhance the overall clarity and accessibility of the paper.

2. I understand that the setup in this paper seems different from previous work called MI-PLL, but the motivation of this pape is that earlier "MI-PLL" (which has a different meaning than this paper) methods cannot handling the long-tail issue well. What’s puzzling is that it doesn’t actually compare the proposed approaches to those prior MI-PLL methods in the experiments, which feels really inconsistent.

3. From the algorithmic side, I’m left wondering why the proposed methods should work for long-tail distributions. The explanation for how they handle rare classes is pretty vague, and it’s not easy to see how the approach directly addresses the long-tail problem. And the assumption that transition function is known to the learner is too strong. As far as I know, in weakly supervised learning people generally don't make such strong assumption, but it is crucial to learn the function。

**Questions:**

Please see above

---

> ### Author Response · Authors · 2024-11-21
>
> > Weakness 1
>
> We have simplified the notation per your request. In the revised version we have uploaded, we set $\mathcal{Y} = [c]$ and got rid of both $l_j$ for denoting labels and $r_{l_j}$ for denoting their ratios.
> We also (i) included tables summarizing our notation (please see Table 7 and 8 in the appendix), and (ii) simplified part of the notation/added further explanations.
>
> > Weakness 2
>
> Because of the popularity of neurosymbolic learning and its rich literature, our experiments focus on neurosymbolic learning scenarios where MI-PLL arises. In neurosymbolic learning, the SOTA approach to train deep classifiers is the semantic loss (SL) by (Xu et al., 2018). SL has been effective in many tasks, including visual QA (Huang et al., 2021), video-text retrieval (Li et al., 2023b), and fine-tuning language models (Zhang et al., 2023; Li et al., 2024).
>
> Due to its effectiveness, SL is adopted by the most well-known neurosymbolic engines, DeepProbLog (Manhaeve et al., 2018), the successor of DeepProbLog presented in [1], and Scallop (Huang et al., 2021; Li et al., 2023b; et al., 2024).
>
> The reason why we use only the Scallop is due to scalability restrictions: Scallop is the only engine at the moment offering a scalable SL implementation that supports the MAX-$M$ scenario for $M \ge 3$, SUM-$M$ for $M \ge 3$, and the HWF-$M$ for $M \ge 5$. The scalability issues are because we need to compute the candidate label vectors in $\sigma^{-1}(s)$ for a partial label $s$ during training. Notice that computing $\sigma^{-1}(s)$ for training is generally required by neurosymbolic learning. This computation can become a bottleneck when the space of candidate label vectors grows exponentially, as in MAX-$M$.
>
> As empirically shown by (Tsamoura et al., 2021; Wang et al., 2023b), the neurosymbolic techniques from (Manhaeve et al., 2018; Dai et al., 2019; Li et al., 2023a) and [1,2] either time out after several hours or lead to very low accuracy. In fact, Scallop was the only one to lead to good accuracy for $M > 3$, with all the other techniques above leading to considerably low accuracy, see Section 6 by (Wang et al., 2023b).
>
> We clarified the above in the revised version, Appendix F.
>
> [1]  Robin Manhaeve, Giuseppe Marra, and Luc De Raedt. Approximate Inference for Neural Probabilistic Logic Programming.
>
> [2]  Zhun Yang, Adam Ishay, and Joohyung Lee. NeurASP: Embracing neural networks into answer set programming.
>
> > Weakness 3: why the proposed methods work for long-tail distributions
>
> As stated at the beginning of Section 4, our algorithms mitigate imbalances by enforcing the class priors ($\mathbf{r}$) to a classifier’s predictions. This idea is commonly applied in long-tail learning. For example, in supervised learning, logit adjustment (Menon et al., 2021) increases the frequency of predicting long tail labels via $\mathbf{r}$. For standard PLL, SoLar (Wang et al., 2022) finds pseudo-labels by matching the classifier’s predictions towards the class priors $\mathbf{r}$.
>
> The intuition behind our algorithms and the above techniques is that $f$ tends to predict the labels that appear more often in the training data. Enforcing $\mathbf{r}$ gives more importance to the minority classes at training time (as in ILP) and encourages $f$ to predict minority classes at testing time (as in CAROT).
>
> Our empirical analysis validates the above intuition.  For example, on the Smallest Parent task for $\rho= 50$.
> We see that the accuracy for the last three classes for which we have very few training data is improved by our training time imbalance mitigation (row ILP(ALG1) for $\rho= 50$ in Table 2) dramatically, e.g., for the seventh class, the accuracy increases from 0. to 38.30%, while for the ninth class, from 3.33% to 40.00%.
>
> We added the corresponding plots in Figure 8 in the appendix of the revised submission.
>
> > Weakness 3: the assumption that transition function is known is too strong
>
> First, we point out that in the literature of weakly supervised learning and PLL, the assumption that the transition is fully known to the learner is commonly made, such as in [3, 4].
>
> Second, this assumption is also realistic in our setting. As we noted, MI-PLL arises in the neurosymbolic settings in which some background knowledge is available -- this background knowledge corresponds to $\sigma$ in our formulation -- and the objective is to use this knowledge to build deep models that are of smaller size or are more accurate. For example, by exploiting a few simple symbolic rules, the authors of (Huang et al., 2021) managed to build neurosymbolic architectures for visual question answering that had substantially higher accuracy than state-of-the-art deep networks for that task, see Table 2 from (Huang et al., 2021).
>
> [3] Aditi Raghunathan, Roy Frostig, John Duchi, and Percy Liang. Estimation from indirect supervision
> with linear moments.
>
> [4] Brendan van Rooyen and Robert C. Williamson. A theory of learning with corrupted labels.

---

### Official Review · Reviewer_KhYb · 2024-11-02

**Soundness:** 2
**Presentation:** 2
**Contribution:** 3
**Rating:** 5
**Confidence:** 3

**Summary:**

In the multi-Instance partial label learning problem, the authors provide class-specific error bounds and show that the transition function $\simga$ can significantly impact learning imbalances. On the practical side, we first propose a statistically consistent technique for estimating the marginal of the hidden labels given partial labels. They propose two algorithms that mitigate imbalances at training- and testing-time. The first algorithm assigns pseudo-labels to training data based on a novel linear programming formulation of MI-PLL. The second algorithm uses the hidden label marginals to constrain the model’s prediction on testing data

**Strengths:**

1. They derive class-specific error bounds that depend on the MI-PLL risk under minimal assumptions.
2. They introduce an algorithm to estimate the marginal of the hidden labels.
3. They mitigate learning imbalances at testing-time by modifying the model’s scores to adhere to the estimated ratios.

**Weaknesses:**

1. The presentation about the algorithm in Section 4.2 becomes confusing from line 300. I will list my doubts in the questions below.
2. Can you explain why you enforce the first three constraints in Equ. (6)?
3. In experiments, for $\rho = 0$, i.e., the hidden label distribution is unmodified and balanced, the performance of CAROT is even worse than LA under $M=5$.
4. Figure 1 seems to indicate that within 100 epochs, almost all classes are not well classified. Does the model converge in 100 epochs? If not, can you provide the curve after 100 epochs until the network converges?

**Questions:**

a).	In line 301, why is each training sample uniquely associated with a Boolean formula of $R_{\ell}$ disjuncts of the form $\Phi_{\ell} := \psi_{\ell, 1} \vee \ldots \vee \psi_{\ell, R_n}$? Is each $\psi_{i, t}$ a bool variable or a function? How to determine $R_{\ell}$.

b).	In line 302, why it is that “Formula $\Phi_{\ell}$ encodes the valid label assignments for the $\ell$-th partial training sample subject to $\sigma$ and $s_{\ell}$?

c).	Can you show that “Each disjunct in $\Phi_{\ell}$ is a conjunction of Boolean variables from $\\{ q\_{\ell,i,j} \\}\_{i,j}$, where $q_{\ell,i,j}$ is true if and only if $x_{\ell, i}$ is assigned to the $j$-th label in $\mathcal{Y}$”.

d)    The introduction of $\alpha_{\ell, t}$ is also not easy-to-follow.

e)   Exact definition of $\Phi_{\sigma}$ in Section 4.1 is missing.

---

> ### Author Response · Authors · 2024-11-21
>
> > Weakness 1
>
> To improve readability, we uploaded a revised version of our submission in which we (i) included an example of the linear program formulation (6) -- please see Appendix D.3-- (ii) summarized our notation in Table 7 and 8, and (iii) rewrote parts of Section 4.1 and 4.2.
>
> >  Weakness 2
>
> The first three constraints are because we employed the Tseytin transformation in conjunction with the work in (Srikumar & Roth, 2023) to construct the linear program. Further details are given in the answers to your Questions (a)--(d). We added a detailed example demonstrating the Tseytin transformation and the corresponding linear program in Appendix D, as well as we rewrote parts of Section 4.2 to clarify several points.
>
> >  Weakness 3
>
> Upon review, we found that, for $\rho = 0$ and $M = 5$, CAROT still outperformed LA when the model was trained using the standard Semantic Loss (Xu et al., 2018; Huang et al., 2021)) and RECORDS (Hong et al., 2023) -- though LA performed better when applied in conjunction with our linear programming technique.
> In particular, for this scenario,
> the accuracy of SL + LA is 59.88, while
> the accuracy of SL + CAROT is 60.26.
> Also, the accuracy of RECORDS + LA is 59.28, while the accuracy of RECORDS + CAROT is 60.45.
> Our experiments, thus, still demonstrate the meaningful benefits of CAROT in these scenarios.
>
> >  Weakness 4
>
> As we stated in the caption of Figure 1, learning converges in 100 epochs. To clarify, this is the accuracy when training the MNIST classifier in Example 1.1 using the baseline semantic loss (SL) by (Xu et al., 2018) using its scalable implementation by the Scallop engine (Huang et al., 2021).
>
> In Section 5 and Appendix F, we conducted several experiments using the MAX-M scenario (this scenario is also demonstrated in Example 1.1 and Figure 1) for larger $M$'s and compared the accuracy of the baseline methods and our imbalance mitigation techniques.
> For example, for $M=5$ and $\rho=50$, the hidden labels are highly imbalanced. In this case, the baseline model (row SL in Table 1, for $M=5$ and $\rho=50$) has the following class-specific classification accuracies: 0%, 0%, 13.33%, 0%, 0%, 0%, 0%, 0%, 99.28%, 93.35%.
> In contrast, our training-time mitigation technique (row ILP(ALG1) in Table 1) leads to the following class-specific classification accuracies: 0\%, 0\%, 43,33%, 0%, 0%, 0%, 2.29%, 42.80%, 97.61%, 96.91%, substantially improving the classification accuracy for the MNIST digits 2, 6, and 7 -- especially for 6 and 7, the baseline model has accuracy 0.
>
> We added the corresponding plots in Figure 8 of the revised version.
>
> > Question a
>
> The revised version clarifies these points and explained the notation in Table 8. We also demonstrated these notions via an example in Appendix D.
> For the record, mapping partial labels to DNF formulas is standard in the neurosymbolic literature (Xu et al., 2018; Tsamoura et al., 2021; Huang et al., 2021). Below we briefly answer the raised questions:
>
> - $\Phi_\ell$ encodes all the labels vectors $\mathbf{y} \in \sigma^{-1}(s_\ell)$.
>
> - $\phi_{\ell,t}$ is a conjunction of Boolean variables and corresponds to a label vector $\mathbf{y} \in \sigma^{-1}(s_\ell)$ -- we added the definition of $\sigma^{-1}(\cdot)$ at the beginning of Section 4.1 of the revised manuscript.
>
> - $R_\ell$ is the number of label vectors in $\sigma^{-1}(s_\ell)$
>
> > Question b & c
>
> These are by construction as explained in the revised manuscript, please also see our answer to Question (a).
>
> > Question d
>
> As we wrote in Section 4.2 of the original submission, these variables are introduced due to the Tseytin transformation (Tseitin, 1983). The Tseytin transformation converts the $\Phi_\ell$'s from DNF into CNF in polynomial time. Due to lack of space, the detailed derivation of program (6) and a further discussion of why these variables are introduced are in Appendix D.
> We added more explanations in appendix D of the revised version.
>
> > Question e
>
> We clarified the above in Section 4.1 in the uploaded version of our submission. We quickly reply to your question below:
>
> $\Psi_{\sigma}$ maps each hidden class prior $r_j$, $j \in [c]$, to its solution in the following system of polynomial equations:
>
> $[\sum_{(y_1, \dots, y_M) \in \sigma^{-1}(a_j)}\prod_{i=1}^Mr_{y_i}]= [p_1,\dots,p_{c_S}]$, assuming a given $\mathbf{p}$.
>
> Above, $c_S$ is the number of partial labels, $p_j$ is the probability of occurrence of the $j$-th partial label in the training data – this can be observed from the PLL training samples – and  $r_{y_i}$ is the (unknown) occurrence probability of the class $y_i \in \mathcal{Y}$ in the training data --recall that in MI-PLL the gold labels are unknown during training.
>
> In terms of Example 4.1, $\Psi_{\sigma}$ maps each $r_i$, for ${i \in \{0,\dots,9\}}$, to its solution in the following system of polynomial equations:
>
> ${[r_0^2,r_1^2 + 2r_0r_1,\dots,r_9^2 + 2\sum_{i=0}^8 r_ir_9]^\top
>         = [1/10,1/10,\dots,1/10]^\top}$.

---

> ### Comment · Reviewer_KhYb · 2024-12-03
>
> Thank you for your response. I appreciate the clarifications provided. However, I still have some concerns:
>
> 1. The paper uses numerous notations, which are distributed throughout Sections 2–4. This makes it difficult to follow and track all the introduced terms.
>
> 2. In Figure 1, the class-specific accuracy for classes 0, 1, 3, 4, and 5 is shown to be zero. This outcome is particularly concerning in an imbalanced classification setting. For the imbalanced scenario, the balanced error rate is a more appropriate evaluation metric than overall accuracy. Considering the balanced error rate, the performance appears to be unsatisfactory, resulting in a $100\\%$ error rate.

---

> ### Author Response · Authors · 2024-12-03
>
> > The paper uses numerous notations, which are distributed throughout Sections 2–4. This makes it difficult to follow and track all the introduced terms.
>
> This notation is needed for our theoretical analysis in Section 3, as well as for formalizing our algorithms in Section 4.
>
> To ease the presentation and as we already wrote in our previous comments, **we added two detailed notation tables in the appendix that summarize all our notation, see Tables 7 and 8 in the revised manuscript**.
>
> > In Figure 1, the class-specific accuracy for classes 0, 1, 3, 4, and 5 is shown to be zero. This outcome is particularly concerning in an imbalanced classification setting. For the imbalanced scenario, the balanced error rate is a more appropriate evaluation metric than overall accuracy. Considering the balanced error rate, the performance appears to be unsatisfactory, effectively resulting in a $100%$ error rate.
>
> As we already wrote in our earlier responses, **Figure 1 shows the class-specific accuracies obtained by state-of-the-art MI-PLL techniques (namely (Xu et al., 2018; Huang et al., 2021)) under imbalanced scenarios**. Figure 1 does not show the corresponding overall accuracy of the classifier. Figure 8 shows concretely how our imbalance mitigation techniques improve the class-specific accuracies in the MAX-5 scenario for $\rho=50$. The resulting overall accuracies for MAX-$M$ are shown in Table 1.
>
> To ensure a fair comparison and as the reviewer also suggests, the accuracies in Tables 1, 2, 3, 4, 5, and 6 are balanced accuracies, i.e., they are the weighted sums of the class-specific accuracies, where each class-specific accuracy is weighted by the frequency of occurrence of the corresponding class in the testing data.

---

### Official Review · Reviewer_9Ewx · 2024-11-04

**Soundness:** 3
**Presentation:** 2
**Contribution:** 3
**Rating:** 6
**Confidence:** 2

**Summary:**

This paper addresses Multi-Instance Partial Label Learning (MI-PLL). The primary contributions are :  1. a theoretical characterization of class-specific learning imbalances in MI-PLL; 2. practical algorithms for mitigating these imbalances at training and testing stages.

**Strengths:**

The exploration of class-specific risk in MI-PLL settings is a novel angel. The theory is sound and robust. The proposed algorithms lead to empirical improvements.

**Weaknesses:**

I'm a bit concerned on the assumption that the transition $\sigma$ is instance-independent. I'm not sure how consistent it is with the real-world datasets. I think recent literatures on learning with noise and domain changes assume instance-dependent transitions (e.g. https://www.arxiv.org/pdf/2408.16189).

**Questions:**

How hard is it to generalize the theory to allow transition $\sigma$ to be instance-dependent and how will that change the analysis? I think the proofs in Patrini et al., 2017 can be naturally adapted to instance-dependent noise by letting noise transitions $T$ depend on $x$.

---

> ### Author Response · Authors · 2024-11-21
>
> > I'm a bit concerned on the assumption that the transition is instance-independent.
>
> Thank you for the comment.
> We would like to stress that while this assumption -- of $\sigma$ being instance-dependent-- is met in standard PLL and learning via noisy data, it is **not** common in MI-PLL and especially in the neurosymbolic architectures studied in (Manhaeve et al., 2018; Wang et al., 2019b; Dai et al., 2019; Tsamoura et al., 2021; Huang et al., 2021; Li et al., 2023a) that inspired both our work and (Wang et al., 2023b).
>
> This is because of the real-world scenarios where standard PLL (Cour et al., 2011) and MI-PLL arise. A real-world scenario that gives rise to PLL is using multiple imperfect annotators to label the same input instance $x$. In this scenario, as in the setting of learning under noisy labels, it is reasonable to assume that $\sigma$ is instance-dependent, as some instances may be ``noisier" than others, e.g., the annotators may make more mistakes on those instances.
>
> In contrast, MI-PLL is met in neurosymbolic and constrained learning scenarios where the same function, e.g., the same logical theory or the same set of constraints, is used to draw inferences over the predictions of a deep network. Such an example is shown in Figure 2 by the Scallop authors (Huang et al., 2021), where a **common** set of commonsense symbolic rules (corresponding to $\sigma$ in our paper) is used to draw logical inferences over the softmax predictions of a deep classifier $f$ when $f$ is given an image/set of bounding boxes.
>
> > How hard is it to generalize the theory to allow transition to be instance-dependent and how will that change the analysis?
>
> It would be straightforward to extend our theory to the instance-dependent transitions.
> We can do this by dividing the space of inputs into several sub-regions, where in each region, the transition function is a fixed function.
> Then, the instance-dependent $\sigma$ can be viewed as the concatenation of a set of different $\sigma_i$'s for different regions of the input space.
> Since the transition within each sub-region is effectively instance-independent, we can derive generalization bounds for each region by applying the same reasoning used in our original theory.
> The overall generalization bound can be yielded by taking an averaged sum over these bounds at those sub-regions.

---

> ### Author Response · Authors · 2024-11-24
> **More details on instance-dependent transition**
>
> We want to add a few more clarifications regarding the comment
>
> > How hard is it to generalize the theory to allow transition to be instance-dependent and how will that change the analysis?
>
> via Example 1.1, where the training samples are of the form ${(x_1,x_2,s)}$, where $x_1$ and $x_2$ are MNIST digits and $s$ is the maximum of their gold labels, i.e., ${s = \sigma(y_1,y_2) = \max\lbrace y_1,y_2\rbrace}$ with $y_i$ being the label of $x_i$.
>
> First, consider two training samples ${(x_{1,1}, x_{1,2}, s_1=1)}$ and ${(x_{2,1}, x_{2,2}, s_2=1)}$.
> When $\sigma$ is instance-independent, then for a given partial label $s$, the set $\sigma^{-1}(s)$ of all candidate label vectors $\mathbf{y}$ for which $\sigma(\mathbf{y})=s$, are independent of the input instances. In terms of Example 1.1, the above means that:
> $\sigma^{-1}(s_1=1) = \sigma^{-1}(s_2=1) = {\{(0,1),(1,0),(1,1)\}}$.
>
> Now, let's consider an instance-dependent variant of this problem. Suppose that $s_2$ is generated by another transition function so that the gold label of $x_{2,2}$ cannot be zero, i.e., the valid candidate label vectors for the second training sample are $\{(0,1), (1,1)\}$. In this way, the above means that $\sigma^{-1}(s_1=1) \neq \sigma^{-1}(s_2=1)$.
>
> To reduce this problem to an **instance-independent** one, we can define a new transition function $\sigma’$ so that the new partial label is a tuple of the form (original partial label, transition identifier), where the identifier indicates which transition function is used to produce the original partial label. In this way, for the above example, the training samples are ${(x_{1,1}, x_{1,2}, (s_1=1, id_1=1))}$ and ${(x_{2,1}, x_{2,2}, (s_2=1, id_2=2))}$. This is essentially the process of “dividing the space of inputs into several sub-regions, where in each region, the transition function is a fixed function”, as we wrote in our earlier reply.
>
> Notice that since the transition within each sub-region is effectively instance-independent, we can derive generalization bounds for each region by applying the same reasoning used in our original theory. The overall generalization bound can be yielded by taking an averaged sum over these bounds at those sub-regions.

---

### Author Response · Authors · 2024-11-21

We would like to sincerely thank the reviewers for their constructive feedback and valuable suggestions, which have helped us improve the quality of our manuscript.

In our responses below, we have carefully answered all the questions and weaknesses that are raised by the reviewers. To better address those concerns, we have also uploaded a **revised version of our manuscript** accordingly, with all changes clearly marked in ${\color{red} \textbf{red color}}$ for ease of reference.

The changes in the revised version include:
- Simplified notations, with a comprehensive notation table in Table 7&8.
- Additional explanation to the construction of Equation (1)
- Additional explanation to CAROT and the function $\Psi_\sigma$ (Section 4.1)
- Additional explanation to the training-time imbalance mitigation algorithm (Section 4.2) and the derivation of the linear programs with examples (in Appendix D.2)
- Explanation of why we consider semantic loss and Scallop, in Appendix F
- Additional plots of experiment results to demonstrate the benefits of considering the label priors, in Figure 8

Thank you for your time and consideration!

---

### Author Response · Authors · 2024-12-04
**Summary of Reviewer's Questions and Our Responses (as of 12/03)**

Dear Reviewers and ACs,

We want to summarize our responses to the reviewers’ main comments, along with the updates we have incorporated into our revised submission:

-    **Regarding extending MI-PLL and our theoretical analysis to instance-dependent transitions (Reviewer 9Ewx)**, we showed that is possible, providing also a concrete example to ground the discussion.

-    **About comparing against other neurosymbolic baselines (Reviewer 98Rq)**, we considered the Scallop engine by (Huang et al., 2021), as it is the only one that can scale to our scenarios in Section 5 and Appendix F. Previous work already demonstrated that other neurosymbolic engines scale poorly to large scenarios as ours (Tsamoura et al., 2021; Wang et al., 2023b). We included this discussion in Appendix F in our revised manuscript.

-   **Regarding why our techniques handle long-tail distributions (Reviewers 98Rq and QVdr)**, lines 243—246 in our original submission describe the intuition: enforcing the class priors gives more importance to the minority classes at training time (as ILP does in Section 4.2) and encourages $f$ to predict minority classes at testing time (as CAROT does in Section 4.3). Furthermore, our experiments in Section 5 show that our imbalance mitigation techniques improve over the baselines by up to 14\%. Finally, Figure 8 clearly shows the effects of our imbalance mitigation techniques over the baselines in our experimental scenarios.

-  **About the comment that our assumption that the transition function is known is too strong (Reviewer 98Rq)**, this assumption is common in the literature of weakly supervised learning and partial label learning (Raghunathan et al., 2016) and (van Rooyen and Williamson et al., 2018). Furthermore, this assumption is also made in neurosymbolic learning (Manhaeve et al., 2018; Wang et al., 2019b; Dai et al., 2019; Tsamoura et al., 2021; Huang et al., 2021; Li et al., 2023a).

-   **Regarding applications of MI-PLL (Reviewer QVdr)**, our introduction provides multiple references and application scenarios, e.g., visual question answering and fine-tuning language models. Further, MI-PLL encompasses partial label learning (Cour et al., 2011; Cabannes et al., 2020) and latent structural learning (Steinhardt \& Liang, 2015; Raghunathan et al., 2016).

-   **Regarding the utility of our theoretical results (Reviewer QVdr)**, our results suggest a phenomenon nobody else has observed in the relevant literature -- that learning in MI-PLL is inherently imbalanced -- and motivated us to develop new techniques for imbalance-aware learning. Our error bounds also extend previous relevant results (Cour et al., 2011; Wang et al., 2023b).

-   **About the comment that existing techniques for addressing long-tailed learning could be applicable in MI-PLL (Reviewer QVdr)**, we discuss in lines 076—083, lines 287—294, Appendix D, and Appendix E of our original submission why it is not straightforward to apply existing long-tail learning techniques to MI-PLL. For example, about mitigating imbalances at training time, differently for existing techniques in supervised and semi-supervised learning, in MI-PLL (1) training samples are $M$-ary tuples of instances of the form ${(x_1,\dots,x_M)}$ and (2) the predictions must abide by the constraints coming from $\sigma$. In addition, there was no previous technique for estimating the marginal of the hidden labels $\mathbf{r}$ from MI-PLL training samples (we are the first to provide such a theoretically grounded technique in Section 4.1). Lastly, **our experiments in Section 5 and Appendix F show that a straightforward application of state-of-the-art techniques to MI-PLL may harm the accuracy of the baseline model.**

-   **Regarding our linear program formulation in Section 4.2 and our notation in Section 4.1 (Reviewer KhYb)**, we rewrote Sections 4.1, 4.2, and Appendix D to address these questions, giving in Appendix D an example of our linear program formulation.

-   **About providing a notation table and simplifying our notation (Reviewers 98Rq and KhYb)**, we addressed both comments in our revised paper and added detailed notation tables (Tables 7 and 8) for clarity.

-   **Regarding the accuracies reported in our experimental results (Reviewer KhYb)**, Tables 1--6 show balanced accuracies, i.e., the reported numbers are the weighted sums of the class-specific accuracies, where each class-specific accuracy is weighted by the frequency of occurrence of the corresponding class in the testing data.

**Given that (i) we addressed all of the comments raised by reviewers and (ii) the only comments posted by Reviewer KhYb after our rebuttal had already been addressed in our original rebuttal, it is unclear to us why Reviewers KhYb and 98Rq maintained their original negative scores. Especially Reviewer 98Rq made no comment to our rebuttal.**

Kind regards,
The authors

---

### Meta-Review · Area_Chair_KkTG · 2024-12-23

**Metareview:**

Summary

The paper tackles multi-instance partial label learning within neuro-symbolic learning, redefining imbalance as class-specific error differences rather than instance counts. It offers theoretical insights on how the instance-to-bag label transition function influences this imbalance, alongside a practical method for estimating hidden label marginals.

Strengths

The paper's strengths lie in its novel perspective on class-specific risk, shifting the focus from instance count imbalance to error differences. It provides strong theoretical contributions through class-specific error bounds. Additionally, the proposed algorithm for estimating marginal probabilities of hidden labels introduces practical innovation. Finally, extensive experiments, particularly in neuro-symbolic learning scenarios, showcase the approach's effectiveness.

Weaknesses

The paper has several weaknesses. The assumption that the transition function is known is overly strong; while the authors argue this is common in weakly supervised learning literature, it is more typical to estimate it from clean data rather than assume it is given. Clarity in presentation was initially an issue. Though the authors claim this notation and mathematics issues been addressed in a revised version, the non-traditional multi-instance partial label problem and the non-traditional imbalance problem are not clarified sufficiently in introduction - these terminologies used are against its widely accepted definition and are misleading. Empirical performance improvement appears marginal, with results like an increase from 59.88 (±5.58) to 60.26 suggesting randomness rather than meaningful progress. The scope of the work is narrow, as it does not address classical partial label problems and instead focuses on a specific imbalance definition tied to model performance rather than inherent data characteristics, necessitating stronger arguments for its broader relevance.

Recommendation

The paper is recommended to be rejected due to its overly strong assumption that the transition function is known, marginal empirical, and a narrow scope focused on model performance rather than inherent data issues. These limitations outweigh its theoretical and algorithmic contributions, making its broader relevance and practical impact unclear.

**Additional Comments On Reviewer Discussion:**

The discussion with reviewers led to mixed outcomes, addressing some concerns but leaving critical issues unresolved. Reviewer 9Ewx’s questions were clarified, but their rating remained at 6 with low confidence (2). Reviewer KhYb's concerns about notation and mathematical clarity were resolved, with their confidence to 3, but their performance concerns persisted, keeping their rating at 5. Reviewer 98Rq raised significant questions about the imbalance setup and algorithmic comparison, particularly regarding the transition matrix, which remains unaddressed. Reviewer QVdr maintained a 6 with high confidence but highlighted issues with the partial label definition diverging from existing literature and the disconnect between theory and practice. These unresolved fundamental concerns resulted in a negative decision.

---

### Decision · Program_Chairs · 2025-01-22

Reject